# Pharmacoproteomic characterisation of human colon and rectal cancer

Martin Frejno[1,2] iD, Riccardo Zenezini Chiozzi[2,3], Mathias Wilhelm[2], Heiner Koch[2,4,5], Runsheng Zheng[2] iD, Susan Klaeger[2,4,5], Benjamin Ruprecht[2,6], Chen Meng[2], Karl Kramer[2], Anna Jarzab[2], Stephanie Heinzlmeir[2,4,5], Elaine Johnstone[1], Enric Domingo[1,7], David Kerr[8], Moritz Jesinghaus[9], Julia Slotta-Huspenina[9], Wilko Weichert[9], Stefan Knapp[10], Stephan M Feller[11,12,*] iD & Bernhard Kuster[2,4,6,13,**] iD

## Abstract

Most molecular cancer therapies act on protein targets but data on the proteome status of patients and cellular models for proteome-guided pre-clinical drug sensitivity studies are only beginning to emerge. Here, we profiled the proteomes of 65 colorectal cancer (CRC) cell lines to a depth of > 10,000 proteins using mass spectrometry. Integration with proteomes of 90 CRC patients and matched transcriptomics data defined integrated CRC subtypes, highlighting cell lines representative of each tumour subtype. Modelling the responses of 52 CRC cell lines to 577 drugs as a function of proteome profiles enabled predicting drug sensitivity for cell lines and patients. Among many novel associations, MERTK was identified as a predictive marker for resistance towards MEK1/2 inhibitors and immunohistochemistry of 1,074 CRC tumours confirmed MERTK as a prognostic survival marker. We provide the proteomic and pharmacological data as a resource to the community to, for example, facilitate the design of innovative prospective clinical trials.

**Keywords** CPTAC; CRC65; drug response; patient stratification; proteomics

**Subject Categories** Cancer; Pharmacology & Drug Discovery; Post-translational Modifications, Proteolysis & Proteomics

**Mol Syst Biol. (2017) 13: 951**

## Introduction

Owing to the high molecular heterogeneity of human cancers, profiling technologies such as genomics and transcriptomics have been employed for some time to identify entity-specific molecular subtypes of tumours that can be used for diagnostic refinement, prediction of disease prognosis or to stratify patients for therapy (McDermott *et al*, 2011). More recently, this concept has been extended to the measurement of cancer proteomes and their post-translational modification status as exemplified by the NCI's Clinical Proteomic Tumor Analysis Consortium (CPTAC; Mertins *et al*, 2016; Zhang *et al*, 2016). The first published CPTAC study was on the proteome of colon and rectal cancer (CRC; Zhang *et al*, 2014), for which extensive transcriptional profiling data had previously been used to define consensus molecular subtypes (CMS) of CRC (Guinney *et al*, 2015). The comparison of subtype information at both transcript and proteome level showed general concordance but also significant discrepancies, with many features only detectable at the protein level. In parallel with molecular profiling of cancers, phenotypic drug screening campaigns in large panels of (mostly) genomically well-characterised cancer cell lines representing many tumour entities have been performed (Barretina *et al*, 2012; Medico *et al*, 2015; Iorio *et al*, 2016; Rees *et al*, 2016). These studies aimed at identifying effective drugs or combinations thereof in cellular model systems that recapitulate the genomic alterations found in human tumours and may thus also show efficacy in humans. However, although genomics might play a role in determining drug

1 Department of Oncology, University of Oxford, Oxford, UK
2 Chair of Proteomics and Bioanalytics, Technical University of Munich, Freising, Germany
3 Department of Chemistry, Sapienza – Università di Roma, Rome, Italy
4 German Cancer Consortium (DKTK), Munich, Germany
5 German Cancer Research Center (DKFZ), Heidelberg, Germany
6 Center for Integrated Protein Science (CIPSM), Munich, Germany
7 Wellcome Trust Centre for Human Genetics (WTCHG), University of Oxford, Oxford, UK
8 Nuffield Division of Clinical Laboratory Sciences (NDCLS), University of Oxford, Oxford, UK
9 Institute of Pathology, Technical University of Munich, Munich, Germany
10 Institute of Pharmaceutical Chemistry, Goethe University, Frankfurt am Main, Germany
11 Weatherall Institute of Molecular Medicine, University of Oxford, Oxford, UK
12 Institute of Molecular Medicine, Martin-Luther-University, Halle, Germany
13 Bavarian Biomolecular Mass Spectrometry Center (BayBioMS), Freising, Germany
*Corresponding author. Tel: +49 345 552 2915; E-mail: stephan.feller@uk-halle.de
**Corresponding author. Tel: +49 8161 71 5696; E-mail: kuster@tum.de

sensitivity, given that the majority of drugs act on protein targets, it appears logical to correlate protein expression with drug sensitivity. Recent proteome profiling of the NCI60 cancer cell line panel and a panel of 20 breast cancer cell lines showed that protein signatures predicting drug sensitivity or resistance can be found (Gholami *et al*, 2013; Lawrence *et al*, 2015). Despite these previous efforts, the number of cell lines for any given cancer entity in these panels was limited, impairing the analysis of drug sensitivity for tumour subtypes and possible translation to human patients. Using published transcriptomics data, Medico *et al* (2015) assigned 151 CRC cell lines to different molecular subtypes of CRC patients, in order to identify model systems amenable for drug sensitivity screens in CRC but to date, no comprehensive proteomic dataset on CRC cell lines has been published that would allow for the direct discovery of proteomic signatures of drug sensitivity and resistance in different CRC subtypes.

In this study, we measured the proteomes of a panel of 65 well-characterised human colorectal cancer cell lines (Emaduddin *et al*, 2008) to a depth of > 10,000 proteins and integrated this data with the proteome profiles of 90 CRC patients (Zhang *et al*, 2014) and matched transcriptome profiles (Appendix Supplementary Methods) to define integrated proteomic subtypes of CRC. Integration with drug sensitivity data available for 52 CRC cell lines allowed us to predict the drug sensitivity of cell lines and patients towards 577 drugs or combinations thereof. The analysis revealed that, for example, high MAP2K1 (MEK1) expression renders small-molecule EGFR inhibitors less effective and also identified the kinase MERTK to confer partial resistance to MEK1/2 inhibitors. Analysis of 1,074 CRC patients from the QUASAR 2 trial (Kerr *et al*, 2016) proved that high MERTK expression is a prognostic marker for poor survival, defining this receptor tyrosine kinase as an attractive potential target for therapeutic intervention. We are making the proteomic data and our full analysis available to the scientific community to provide a rich resource for aiding in the design of future prospective clinical studies in CRC based on multi-omics molecular tumour data and phenotypic drug sensitivity data.

# Results

### Proteome profiles of CRC cell lines and patients

We devised a multi-omics data integration strategy to determine integrated proteomic subtypes of human colorectal cancer cell lines and patient samples in order to predict their sensitivity towards a variety of clinical and pre-clinical drugs and combinations thereof (Figs 1A and EV1, Appendix Supplementary Methods). To accomplish this, we first used LC-MS/MS-based shotgun proteomics to measure the proteomes of 65 CRC lines and to quantify their expressed kinomes using Kinobeads (Bantscheff *et al*, 2007; Medard *et al*, 2015). This led to the identification of a total of 11,796 protein groups (median/cell line = 9,447) representing 10,951 genes (median/cell line = 9,068) and 235 human kinases (median/cell line = 155, from Kinobeads experiments; Fig 2A; Table EV1A and B). We next re-analysed the proteomic profiles of 90 CRC patients published by the CPTAC (Zhang *et al*, 2014) using the same analysis pipeline and identified 7,005 protein groups (median/patient = 4,980) representing 6,727 gene groups (median/

patient = 4,901; Fig 2C; Table EV1C, Appendix Supplementary Methods). The CRC65 data contained most proteins of the CPTAC CRC data (Fig 2B), but we noted that gene groups unique to the CPTAC dataset are enriched in extracellular matrix proteins and IgGs as one might expect for tissue samples containing blood vessels and connective tissue. For quantification of full proteomes, we used a modified version of the intensity-based absolute quantitation (iBAQ) approach (Schwanhausser *et al*, 2011) termed gene-centric iBAQ (giBAQ, Appendix Supplementary Methods), while kinomes were quantified using the label-free quantification (LFQ) intensities provided by MaxQuant (Cox *et al*, 2014). For all subsequent data analyses, we used the proteomic data aggregated at the gene group level annotated with gene symbols as identifiers in order to be able to compare proteomics and transcriptomics data (referred to as protein expression/abundance throughout the manuscript). An overview of the data integration pipeline is depicted in Fig EV1.

### Integration of multiple mRNA datasets reveals consensus molecular subtypes of the CRC65 panel

A number of studies have reported CRC patient subtypes based on mRNA measurements, and these were recently consolidated into consensus molecular subtypes (CMS) and applied to patients from the CPTAC study (Guinney *et al*, 2015). On this basis, we sought to determine the CMS membership of CRC cell lines in order to identify cell lines representative of CRC tumour subtypes. We downloaded 10 public mRNA datasets (Appendix Supplementary Methods), eight for the cell lines and two for the patients (Fig 1B), analysed them from the data level closest to raw data available to us and aggregated the datasets using a scheme similar to the one used by Guinney *et al*. This involved the selection of a reference dataset, followed by the selection of a probe set for each gene based on a consistency criterion and subsequent cross-dataset normalisation (Appendix Supplementary Methods). This resulted in a combined expression matrix of 9,737 transcripts (7,113 after filtering for low abundance transcripts) across 145 non-redundant cell lines (including the CRC65 panel) and 89 tumours (Table EV1D). We adapted the single-sample CMS classifier reported by Guinney *et al* to accept gene symbols as identifiers (rather than Entrez IDs; Appendix Supplementary Methods) and predicted the CMS for cell lines and patients based on 382 of the 692 classifier genes contained in the combined expression matrix. The correct classification of 65 out of 81 patients (80%, using the original CMS assignment as the ground truth) provided confidence that cell lines can be placed into CMSs with good accuracy and the resulting subtype labels for the CRC65 cell lines and the CPTAC patients are shown in Fig 1B. A subtype-resolved evaluation of the prediction accuracy using a confusion matrix and a table containing a variety of commonly used metrics for evaluating classification performance can be found in Table EV2E.

### Integrated proteomic subtypes of CRC cell lines and tumours

Despite the fairly deep proteomic measurements, the quantification of proteins across many cell lines (and patients) suffered from an increasing number of missing values for proteins of decreasing abundance (Fig EV2A). We addressed this frequently encountered issue by mRNA-guided and minimum-guided missing value

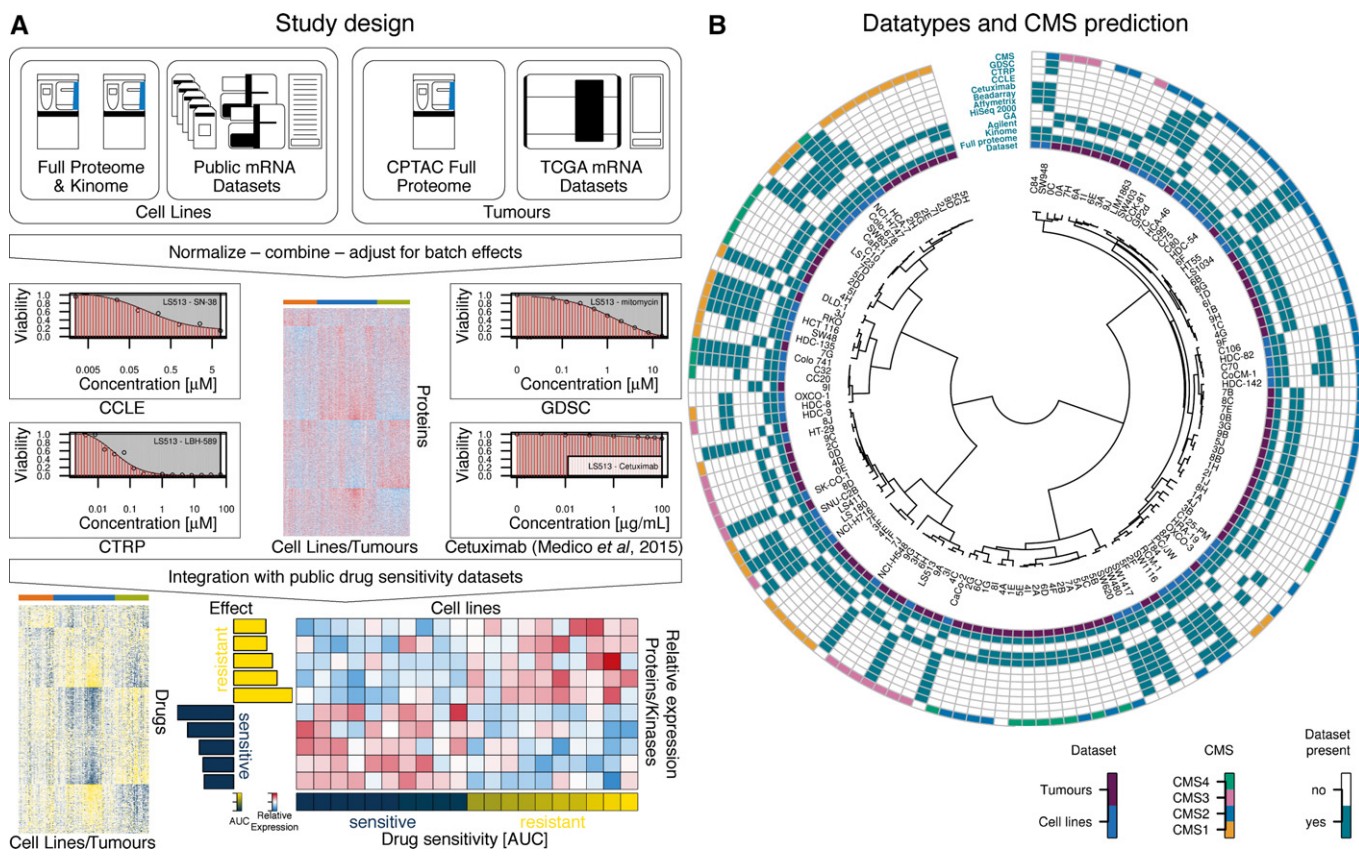

**Figure 1.  Study design, datatypes & CMS prediction for cell lines.**

A   Overview of the multi-omics data integration workflow followed in this study. Proteomic and transcriptomic data generated as part of this study or available from the literature for colorectal cancer (CRC) cell lines and patients were integrated in order to identify cell lines and tumours forming proteomic subtypes. Four published drug sensitivity datasets (abbreviated CCLE, CTRP, GDSC and cetuximab (Medico *et al*, 2015); one dose–response plot for each data source in shown) were overlaid onto the proteomic data to identify protein signatures associated with sensitivity or resistance. An example of an effect-size heat map for one drug, ten proteins and 20 cell lines is shown at the bottom-right (see main text and Appendix Supplementary Methods for details).

B   Circos plot visualising the different datasets that were integrated in this study. The dendrogram in the centre hierarchically ordered tumours (violet "Dataset" label) and cell lines (blue "Dataset" label) based on the mRNA expression of classifier genes from Guinney *et al* (2015). A set of rings around the dendrogram indicates which proteomics data (full proteome, kinome), mRNA technologies (Agilent microarray, Genome-Analyser-based mRNA-Seq, HiSeq-2000-based mRNA-Seq, Affymetrix microarrays or Illumina Beadarrays) and drug sensitivity datasets (cetuximab, CCLE, CTRP or GDSC) were included in this study. The outermost ring indicates the membership of cell lines/tumours in a consensus molecular subtype ("CMS"). Undetermined CMS class labels or unavailable data were left white (see main text and Appendix Supplementary Methods for details). See also Fig EV1.

---

imputation on the peptide level to generate one complete protein expression matrix consisting of 59 cell lines, 81 tumours and 6,254 proteins (Fig EV2, Table EV1E), of which 323 were contained in the CMS classifier by Guinney *et al* (CMSgene in Fig 3A; see Appendix Supplementary Methods for details). In order to estimate protein levels from mRNA levels, we removed systematic differences (Fig EV3A and B) between proteomics and transcriptomics data using MComBat (Stein *et al*, 2015; see Appendix Supplementary Methods). This increased the protein/mRNA correlation for both the CRC65 and CPTAC datasets (Figs EV2B and EV3C), enabling mRNA-guided missing value imputation. After imputing missing values separately for both datasets (Fig EV2D and E, Appendix Supplementary Methods), we accounted for differences in their proteomic depth (Fig EV2C, Appendix Supplementary Methods) using ComBat (Johnson *et al*, 2007) before we merged the two protein expression matrices. Using the combined protein expression

matrix and consensus clustering (Appendix Supplementary Methods), we identified three integrated Full Proteome Subtypes (short: FPSs, namely FPA, FPB and FPC; Fig 3A). Each subtype consisted of cell lines as well as patients in a ratio of 28/34 for FPA, 22/12 for FPB and 9/26 for FPC, indicating that indeed there are cell lines, which are molecularly more similar to tumours than they are to other cell lines. We measured the association of these FPSs with previously published subtypes, as well as genomic and epigenomic features using Fisher's exact test (Table EV2A), and found good overall concordance but with some differences in detail (see discussion). Interestingly, FPA was associated with TP53 mutations, while FPB was associated with ATM mutations, suggesting that p53 signalling in response to DNA damage is perturbed through distinct mechanisms in these two subtypes. FPC showed association with mutations in BCL9L, a transcriptional activator of b-catenin activity, RNF43, an E3 ubiquitin-protein ligase, which acts as a negative

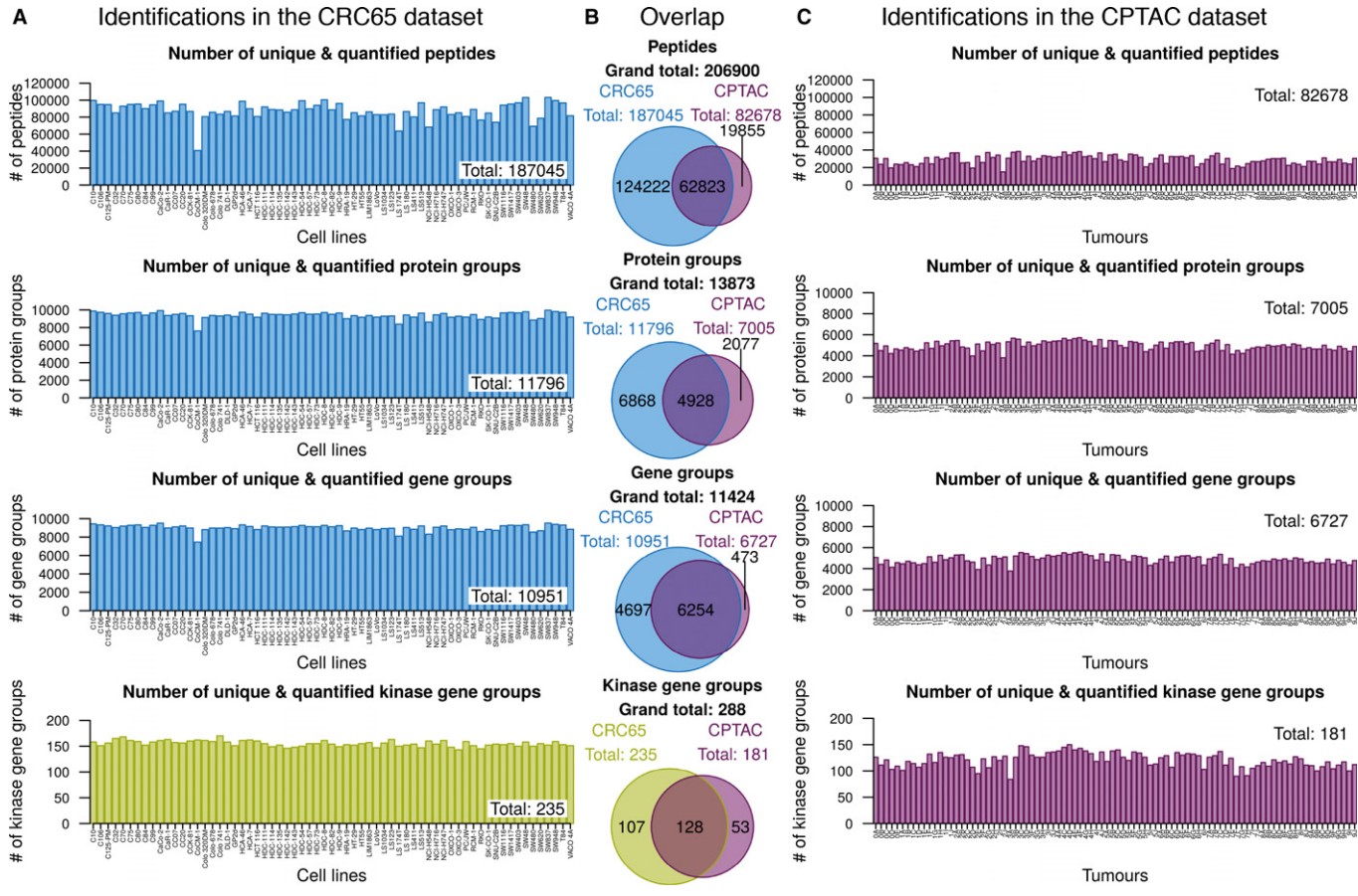

**Figure 2.   LC-MS/MS-based identifications.**

A   Bar charts visualising the number of unique identified and quantified peptides, protein groups and gene groups (full proteomes), as well as kinase gene groups (Kinobeads), across the CRC65 cell line panel (*n* = 65 cell lines). Insets indicate the corresponding number of identifications across the entire dataset.

B   Venn diagrams showing the intersection and complements with respect to identifications in the aforementioned categories across both the CRC65 cell line and CPTAC patient dataset (*n* = 89 tumours). Kinase identifications in the CPTAC dataset were extracted from the full proteome data.

C   Same as (A) for the CPTAC dataset. The different proteomic datasets were colour-coded (green = Kinobeads, blue = CRC65 full proteomes and purple = CPTAC full proteomes).

regulator of Wnt signalling, as well as with mutations in the histone acetyltransferase EP300, all pointing towards deregulated WNT signalling in conjunction with aberrant histone acetylation. Enrichment analysis of functional classifications using MetaCore (Fig EV4, Appendix Supplementary Methods) revealed that proteins significantly under- or overrepresented in the different FPSs fall into specific categories. Briefly, the analysis suggests that FPA is characterised by a "high metabolism, low cell cycle, microsatellite stable (MSI−)" signature, FPB harbours a "high immune response, low metabolism, microsatellite instable (MSI+)" signature and FPC shows a "low immune response, low inflammation, low adhesion (invasion)" signature.

**Integration of drug sensitivity data reveals protein signatures of drug response**

Since the response to drug treatment cannot be determined as quickly and as broadly as would be desirable in clinical studies, we took advantage of the fact that CRC cell lines recapitulate the main

molecular subtypes in CRC and were recently extensively characterised for drug sensitivity as part of major screening efforts. On the basis of the proteomic data, we used elastic net regression (Zou & Hastie, 2005) to predict the response (sensitivity/resistance) of CPTAC patients and CRC65 cell lines to 577 drugs or combinations thereof (Fig 3B, Appendix Supplementary Methods). Briefly, common cell lines (52 cell lines overlap in total; median overlap per drug is 27) between the CRC65 panel and three small-molecule drug sensitivity screens (Barretina *et al*, 2012; Garnett *et al*, 2012; Rees *et al*, 2016) and one study investigating sensitivity of CRC cell lines towards cetuximab (Medico *et al*, 2015) served as the training set (Fig 1, Appendix Supplementary Methods). We used significance analysis of microarrays (SAM; median FDR of 0.001; Appendix Supplementary Methods) to identify drugs which showed specificity for certain FPSs and performed target space enrichment by measuring the association between these subtypes and recurrent annotated targets of these drugs using Fisher's exact test (Fig 3B, Table EV2B). Cetuximab, for example, was more effective in FPA than in the other FPSs, with EGFR target enrichment reaching statistical significance

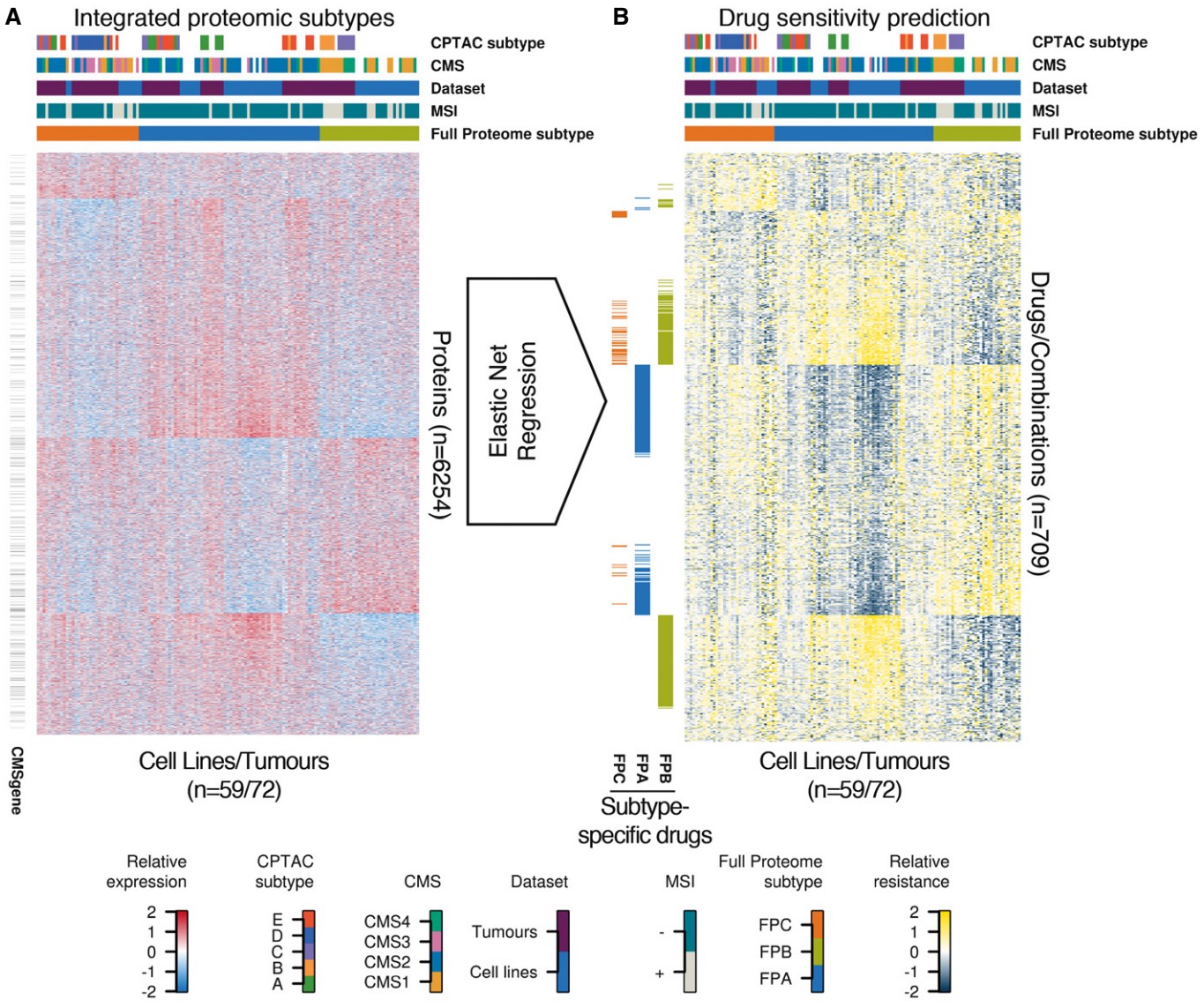

**Figure 3.  From integrated proteomic subtypes to drug sensitivity prediction.**

A   Heat map of standardised, log2-transformed and median-centred giBAQ protein quantification (z-scores) across the combined CRC65/CPTAC dataset. Cell lines/patients are displayed as columns and proteins are shown as rows. Black bars to the left of the heat map indicate the presence of the respective protein in the CMS classifier. Annotation bars on top of the heat map visualise the membership of the different cell lines/patients in five annotation categories.

B   Elastic net regression was used to model drug sensitivity as a function of protein profile with the CCLE, CTRP, GDSC and cetuximab datasets as input. The resulting models were used to predict the drug sensitivity of cell lines/patients (see also Appendix Supplementary Methods). In the heat map of standardised predicted AUC values (z-scores), cell lines/patients are displayed as columns with the same annotations and ordering as in (A), while drugs or combinations thereof are shown as rows. Coloured bars to the left of the heat map indicate drugs, which are more effective in a respective subtype (see main text and Appendix Supplementary Methods for details). See also Figs EV2–EV4.

($P$ = 0.05). FPA was also significantly (multiple-test adjusted $P < 0.05$) associated with drugs targeting the kinases MAP2K1 and MAP2K2 (MEK1 and 2; mainly selumetinib/AZD6244 and its combinations with other compounds), as well as with drugs and combinations targeting the bromodomain containing protein BRDT (mainly by the compound JQ-1), histone deacetylases or HDACs (vorinostat among others) and the nicotinamide phosphoribosyltransferase NAMPT (daporinad among these). Even though more drugs were differentially effective in FPB than in FPA (205 versus 170), most of the former did not show significant enrichment in their target space, making it difficult to link the drug phenotype to the presumed mechanism of action. However, FPB did show positive association with drugs targeting inhibitors of apoptosis (IAPs; all these compounds are mimetics of second mitochondrial activator of caspases, SMAC) as well as negative association with drugs targeting EGFR, while FPC showed association with drugs targeting dihydrofolate reductase (DHFR, including methotrexate). The complete list of predicted drug sensitivities for the patients and cell lines can be found in

Table EV3A, while mean effect sizes of all proteins from elastic net regression can be found in Table EV3B.

## High MERTK expression in CRC cell lines is predictive of resistance to MEK1/2 inhibitors

Since many targeted cancer drugs act on kinases, we performed independent experiments focussing on the expressed kinome of the CRC65 cell line panel to identify kinases associated with drug sensitivity or resistance. Pulling down protein kinases using immobilised kinase inhibitors (Kinobeads; Bantscheff et al, 2007; Medard et al, 2015) led to the reproducible (average $R = 0.91$ between replicates; Fig EV5A) quantification of 138 kinases by mass spectrometry (Appendix Supplementary Methods), which correlated well ($R = 0.94$; Fig EV5B) with data obtained by Western blotting and densitometry for the kinases EPHA4 (Fig EV5C; variable expression across the panel) and ABL1 (Fig EV5D; relatively stable expression across the CRC65 panel). While kinase expression as measured using Kinobeads correlated reasonably well with measurements in full proteomes, an enrichment of kinases was clearly visible in the Kinobeads data (Fig EV5E) and led to the identification and quantification of more than 50 kinases not detected in the deep proteome analysis (Fig EV5F). Consensus clustering identified three Kinobeads Subtypes (KSs) KA, KB and KC (Fig EV5G) that fully recapitulated the Full Proteome Subtypes and showed significant association with other subtype classifications (Table EV2C).

By searching for differentially expressed kinases between the different Kinobeads Subtypes using SAM (median FDR of 0.01), we identified proteins frequently mutated in microsatellite instable (MSI+) tumours with concomitant decrease in expression of proteins such as ACVR2A and TGFBR2 (Kim et al, 2013), as well as the receptor tyrosine kinase EPHA2, which was overexpressed in KC (Table EV4). Overexpression of EPHA2 is known to predict resistance towards cetuximab in CRC (Strimpakos et al, 2013). Elastic net regression was used to identify kinases associated with drug sensitivity or resistance and confirmed the strong association of EPHA2 with resistance towards cetuximab (Fig EV5H, Table EV3C and D). We also noted that overexpression of MAP2K1 (MEK1) was associated with resistance to two-thirds (12/18) of all inhibitors targeting EGFR (Fig 4 for example). Activating mutations in K57 of MAP2K1 were previously shown to be a potential mechanism of primary resistance towards EGFR-targeted therapy (Bertotti et al, 2015), and our data suggest that high expression can have a similar effect. SAM also identified MERTK —a receptor tyrosine kinase correlating with disease progression in melanoma (Schlegel et al, 2013) and not detected in the full proteome measurements—to be differentially expressed between the different Kinobeads Subtypes ($q$-value < 0.01). MERTK is down-regulated in KRAS-mutant CRC (Watanabe et al, 2011) and up-regulated in pancreatic cancer cell lines resistant towards selumetinib (Beech & Kelly, 2014). Interestingly, we also recurrently found high MERTK expression associated with resistance towards MEK1/2 inhibitors as well as other drugs from multiple drug sensitivity datasets (Figs 5A and B, and EV6A, Appendix Supplementary Methods). To validate hypotheses arising from the above predictions, we performed a series of in vitro drug treatment experiments.

On the basis of MERTK expression, we selected three cell lines (CC07, HDC-143 and SK-CO-1) predicted to be sensitive to two MEK1/2 inhibitors (RDEA119 and PD-0325901) as well as three cell lines (C10, CaCo-2 and T84) predicted to be resistant to these drugs (Fig EV6B). Cell viability assays and nuclei counting of cells (Fig 6C and D) showed that the predictions could be confirmed, as cell lines expected to be resistant did not respond as well as cell lines expected to be sensitive ($P \leq 0.05$, one-sided Mann–Whitney test). Similar results were obtained for the predicted association of ACVR2A expression with drug sensitivity to AUY922 (an HSP90 inhibitor) and BAY 61-3606 (a designated SYK inhibitor) in LS-180, OXCO-1 and RKO cells (sensitive) or C125-PM, HDC-111 and HT55 cells (resistant; Fig 5C and D). Since ACVR2A expression is reduced in microsatellite instable CRC, these tumours might benefit from treatment with these drugs.

We next asked whether knockout of MERTK in MEK1/2-inhibitor-resistant C10 cells could re-sensitise them to treatment with RDEA119 (MEK1/2 inhibitor). A Western blot-based validation of the CRISPR/Cas9-mediated knockout is depicted in Fig 5E. We also confirmed the knockout by sequencing of the targeted genomic region, which showed two insertions: one in exon 7 (FN3 domain) and one in exon 14 (kinase domain), inducing frameshifts. As depicted in Fig 5F, C10$^{MERTKN^{KO}}$ was more sensitive to RDEA119 than the parental C10$^{MERTKN^{WT}}$ cell line. We next tested whether the combination of MEK (by RDEA119 or PD-0325901) and MERTK inhibition (UNC569) could re-sensitise MEK1/2-inhibitor-resistant cell lines. As shown in Fig EV6C–F, co-treatment indeed significantly ($P \leq 0.05$, one-sided Mann–Whitney test) reduced the viability of MEK1/2 inhibitor-resistant cell lines to levels comparable to sensitive cell lines. However, this reduction appeared to be mainly due to UNC569 on its own, since both MEK1/2 inhibitor-sensitive and resistant cell lines show comparable response to the co-treatment when treated with UNC569 alone.

## MERTK is a prognostic survival marker in CRC patients

In order to evaluate the general expression status as well as the clinical impact of MERTK expression in CRC, we quantified its abundance in 1,074 patients enrolled in the QUASAR 2 trial by immunohistochemistry (IHC). After establishing cell-based positive (CaCo-2) and negative (RKO) controls with high and low expression of MERTK, respectively (Fig 6A and B), tissue microarrays (TMAs) were stained for MERTK and the fraction of positive cells, as well as the staining intensity, were quantified by pathologists. While MERTK expression was found primarily at the membrane in CaCo-2 cells, the TMA data showed combined cytoplasmic and membrane staining. Therefore, TMAs were categorised to have either high or low MERTK expression (more or < 5% cytoplasm/membrane-positive tumour cells, Appendix Supplementary Methods) without distinguishing between these compartments. Figure 6C shows three representative TMA samples alongside their quantification of MERTK, also highlighting the strong variability of MERTK expression in primary tumours. Modelling the outcome variables of the QUASAR 2 trial as a function of MERTK expression showed that high MERTK expression was prognostic of worse 5-year overall survival (OS), disease-free survival (DFS) and recurrence-free survival (RFS) in both univariate and multivariate Cox proportional hazards regression (Table EV5, Fig 6D). Notably, MERTK expression is more informative in predicting these outcome variables than the patients' treatment status, since its coefficient was significant in the final multivariate Cox proportional hazards model, while the

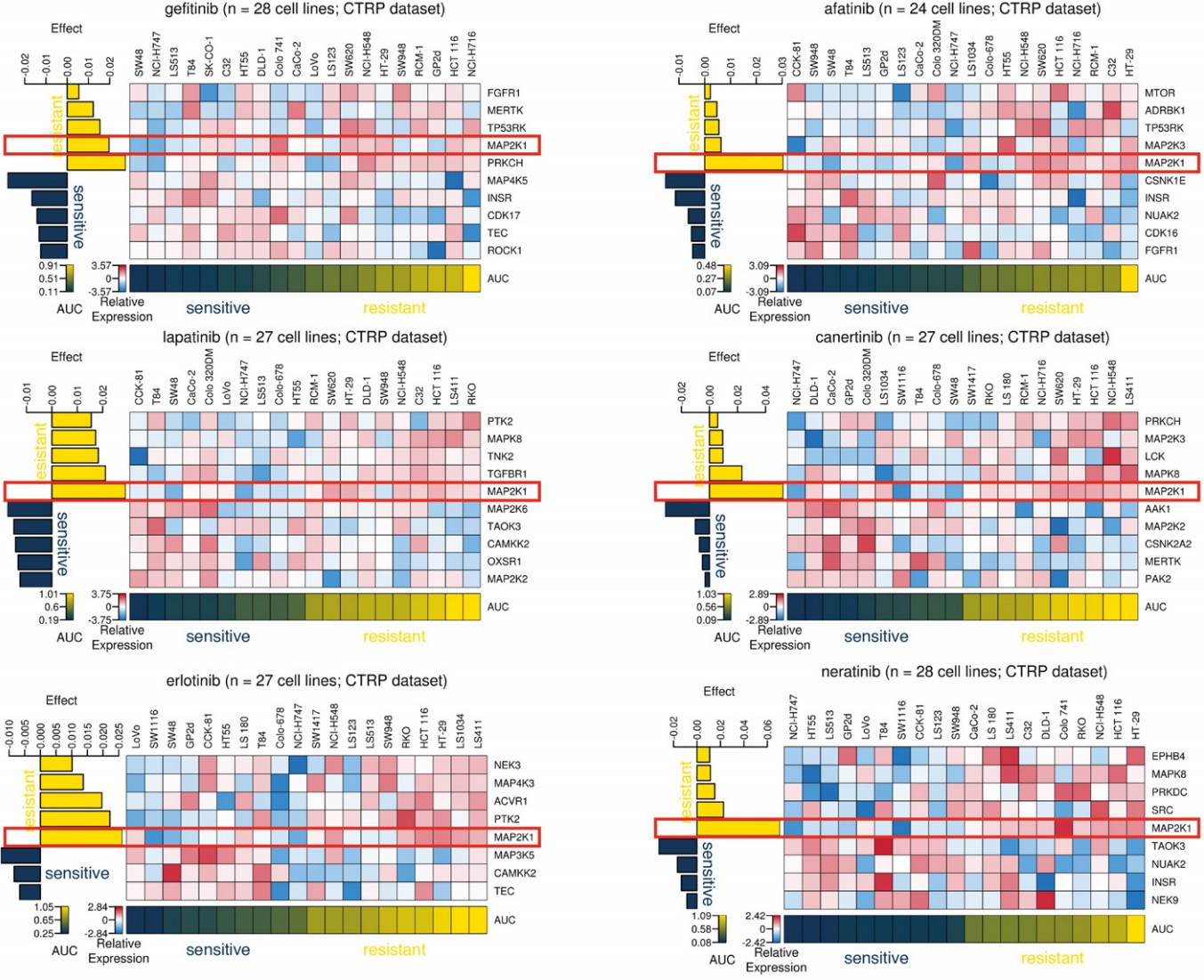

**Figure 4.  MAP2K1 is a predictive marker for inhibitors targeting EGFR.**

Effect-size heat maps of six drugs (see titles of panels) targeting EGFR. It is evident that the different drugs showed different profiles but also that high MAP2K1 expression (blue/red gradient across cell lines) was consistently associated with drug resistance (dark blue/yellow gradient across cell lines; AUC: area under the curve; see main text and Appendix Supplementary Methods for details). See also Fig EV5.

coefficient of the treatment variable was not (see also Appendix Supplementary Methods). Comparing tumours with high versus low MERTK expression resulted in a multivariate hazard ratio (HR) of 1.61 for OS (CI$_{0.95}$ = 1.1–2.36; significant coefficient for MERTK with $P$ = 0.015; Wald test), 1.70 for DFS (CI$_{0.95}$ = 1.24–2.33; significant coefficient for MERTK with $P$ = 0.00094; Wald test) and 1.77 for RFS (CI$_{0.95}$ = 1.26–2.48; significant coefficient for MERTK with $P$ = 0.00087; Wald test), respectively. High expression of MERTK was also significantly associated with BRAF$^{V600E}$ mutations (12% of the CRC65 cell line panel and 13% of the QUASAR 2 cohort carry the mutation, Fig EV5D) and stage T4 tumours, while low cytoplasmic/membranous expression was associated with wild-type BRAF and stage T2/T3 tumours (two-sided Fisher's exact test $P$ < 0.05, Table EV2D). We note that more work is needed in order to determine the direction of causality for these observations and whether

MERTK might qualify as a drug target in CRC in addition to being a prognostic survival marker in CRC patients.

## Discussion

Recent landmark studies established resources like the "Genomics of Drug Sensitivity in Cancer" (GDSC; Iorio *et al*, 2016), the "Cancer Cell Line Encyclopedia" (CCLE; Barretina *et al*, 2012) and the "Cancer Therapeutics Response Portal" (CTRP; Rees *et al*, 2016), all of which focus on the identification of genomic and transcriptomic associations with drug sensitivity. However, since drugs almost always target proteins, it appears obvious to incorporate proteome-wide measurements into drug sensitivity association studies. Here, we overlaid these large-scale drug sensitivity datasets and data on

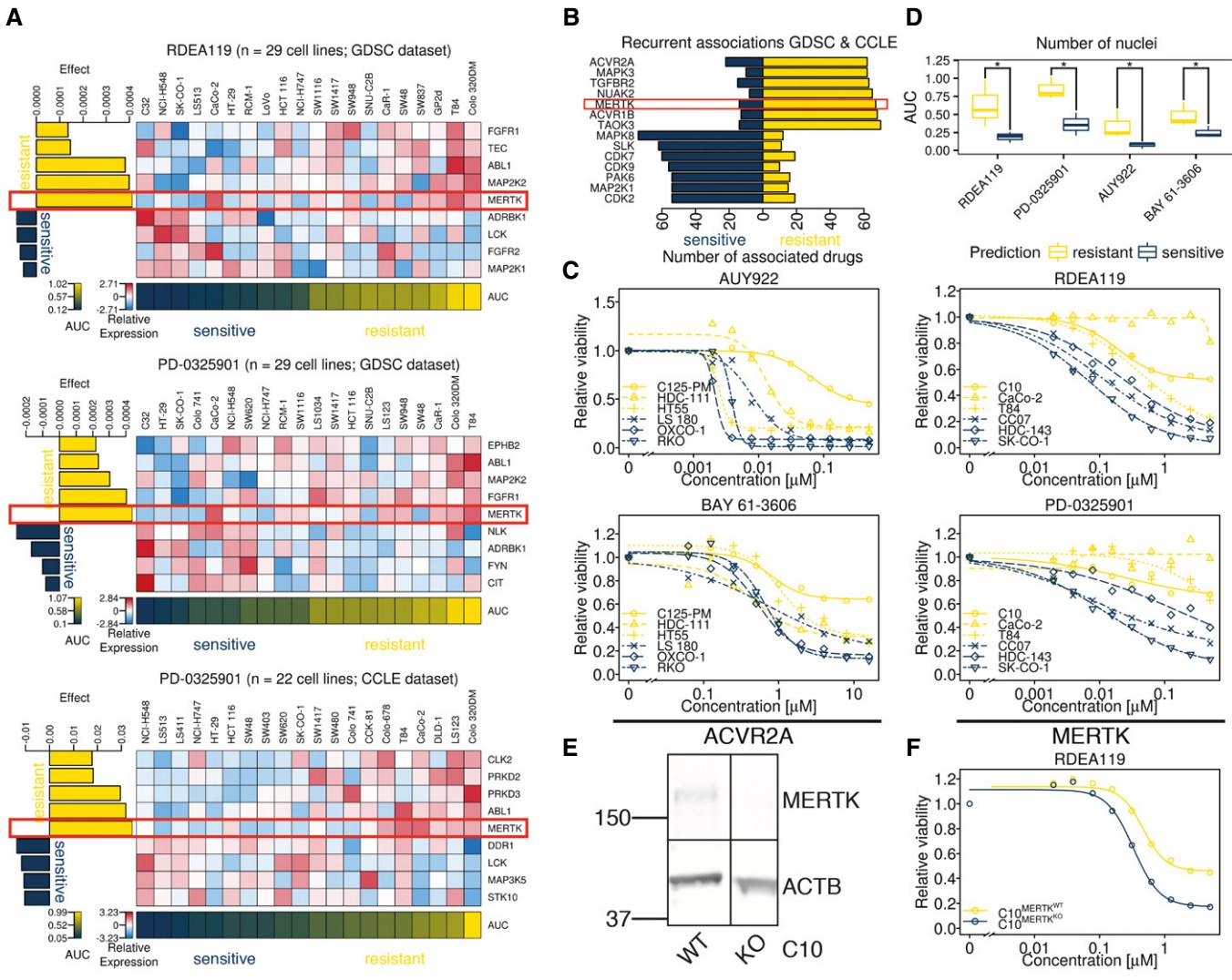

**Figure 5.  MERTK is a predictive marker for inhibitors targeting MEK1/2 in CRC cell lines.**

A   Effect-size heat maps of two drugs (one from two different drug sensitivity screens) targeting MEK1/2 show consistent association of high MERTK expression with drug resistance. The colour scheme is the same as in Fig 4.

B   Bar chart visualising the top kinases recurrently associated (absolute effect size > 0) with drug resistance (top seven bars) and sensitivity (bottom seven bars) in the GDSC and CCLE drug sensitivity datasets.

C   Dose–response curves of two drugs for which high MERTK (left panels) or ACVR2A (right panels) expression was predicted to confer drug resistance. For each drug, three cell lines predicted to be sensitive (dark blue) and three cell lines predicted to be resistant (yellow) were assayed for viability. The experimental data validated that cell lines predicted to be more sensitive to a drug indeed showed this phenotype (data represent the average of three technical replicates; see Appendix Supplementary Methods for details).

D   Boxplots summarising the data shown in panel (C) using the area under the curve (AUC) as a measure for drug sensitivity. The whiskers extend to the minimum and maximum AUC for a given drug and sensitivity prediction, while the median AUC is marked with a bold horizontal line inside a box spanning the interquartile range (IQR) from the 25% quantile (lower horizontal line) to the 75% quantile (upper horizontal line). High AUC values indicate drug resistance. Again, it is evident that cell lines predicted to be more resistant to a certain drug were in fact significantly (*$P \leq 0.05$, one-sided Mann–Whitney test) more resistant than cell lines predicted to be more sensitive.

E   Western blot of C10^MERTKN^WT and C10^MERTKN^KO cells, visualising successful knockout by CRISPR/Cas9.

F   Dose–response curve of RDEA119 (MEK1/2 inhibitor) in C10^MERTKN^WT and C10^MERTKN^KO cells. The knockout is more sensitive to MEK1/2 inhibition than the wild type. See also Fig EV6.

cetuximab (Medico *et al*, 2015) onto extensive proteomic profiles of 65 colon cancer cell lines, created as part of this study and 90 published human colon cancer proteomes (Zhang *et al*, 2014; reprocessed for this study) in order to identify molecular subtypes of colorectal cancer and molecular markers of drug response with translational potential to the human disease.

It was shown before that cell lines included in the CRC65 panel might capture core molecular subtypes of CRC (Mouradov *et al*, 2014; Medico *et al*, 2015); however, the recently established consensus molecular subtype CMS of these cell lines was not known as of yet. By combining and re-analysing multiple public mRNA datasets acquired using a variety of technologies across a number of

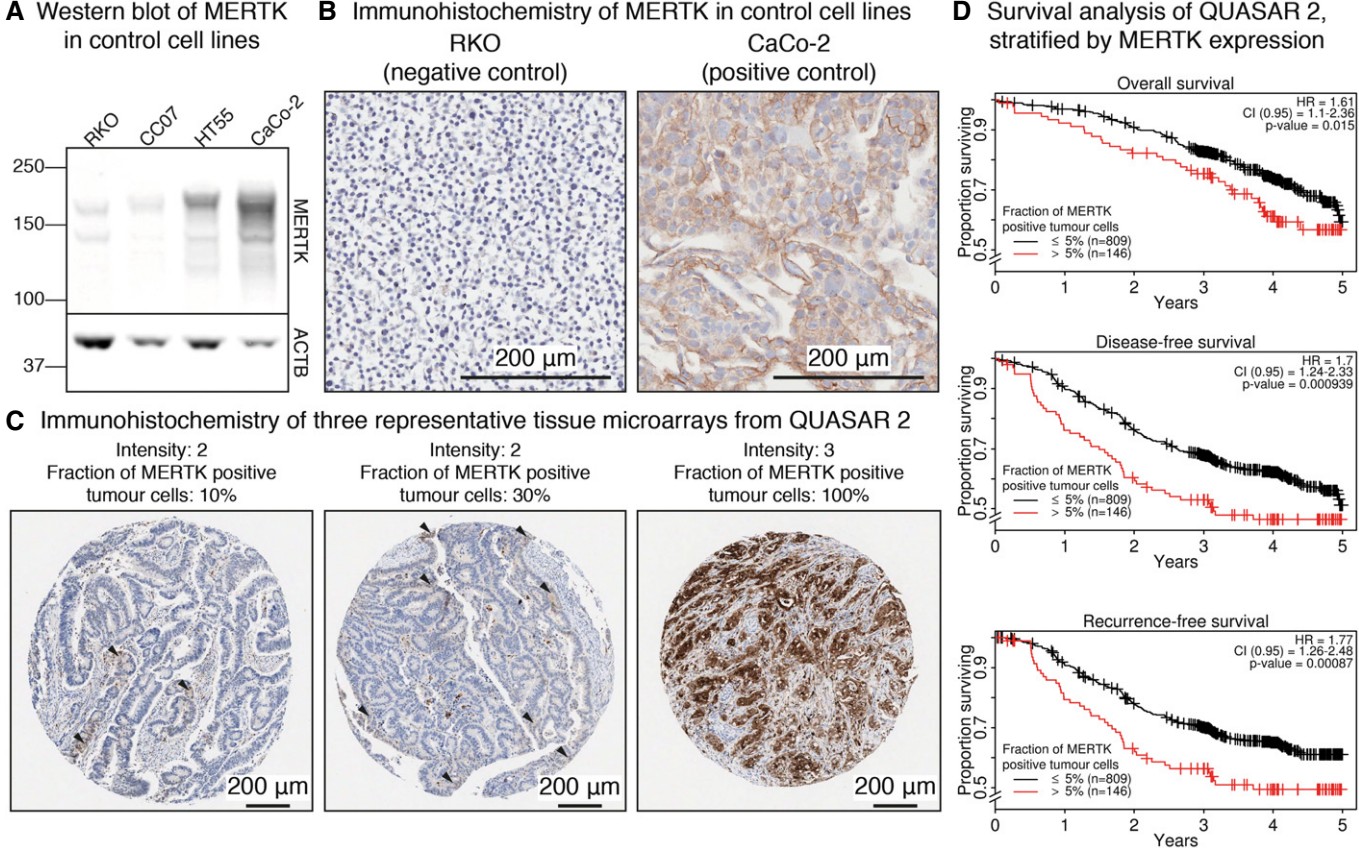

**Figure 6.  MERTK is a prognostic marker in CRC patients.**

A   Western blot visualising MERTK expression in two negative (RKO and CC07) and two positive control cell lines (CaCo-2 and HT55).
B   IHC staining of MERTK expression in RKO (negative control) and CaCo-2 (positive control) cells.
C   IHC staining of three representative TMAs from patients enrolled in the QUASAR 2 clinical trial and the corresponding quantification of the signal by pathologists. Arrowheads indicate tumour cells positive for MERTK.
D   Kaplan–Meier plots showing worse overall, disease-free and recurrence-free survival of patients with high cytoplasmic/membranous expression of MERTK. *P*-values indicate the significance of MERTK as a predictor based on the Wald statistic.

laboratories involving several different batches of cell lines cultivated under a diverse set of conditions, we were able to assign 42 of them to a defined CMS subtype using the single-sample classifier developed by Guinney *et al* (2015), confirming that these cell lines indeed represent the main molecular subtypes of CRC. Dunne *et al* recently noted that patients might be misclassified by the CMS classifier due to regional heterogeneity in tumours arising from stromal contributions to measured gene expression (Dunne, *et al*, 2016), highlighting the need for independent unsupervised class discovery while searching for representative cell lines. This might also explain why the cluster in Fig 1B containing most of the CMS4 tumours did not contain cell lines annotated as CMS4, since cell lines only capture the epithelial component of a tumour. The transcriptional profiles of the cell lines were consistent across the different datasets if systematic differences between them were accounted for.

Combining the CRC65 and CPTAC datasets then enabled the identification of integrated Full Proteome Subtypes of CRC (consisting of cell lines and patients), which were associated with previously published mRNA-based and proteomics-based subtypes. Since the CMS represent the current consensus in the field, we note that they could only

be partially recapitulated from proteomics data. The data analysis grouped CMS1 and CMS4 together into Full Proteome Subtype FPB, which might be explained by the fact that mesenchymal genes over-represented in CMS4 are mainly of stromal rather than epithelial origin and will therefore largely evade detection in cell culture experiments (Calon *et al*, 2015; Isella *et al*, 2015). Despite this loss in granularity, the data identified clear proteomic signatures characterised by "high metabolism, low cell cycle, MSI−" for FPA, "high immune response, low metabolism, MSI+" for FPB and "low immune response, low inflammation, low adhesion (invasive)" for FPC.

Due to the substantial overlap of our CRC65 full proteome data on cell lines with the CPTAC patient data, we were able to predict the drug sensitivity of both cell lines and patients towards 577 drugs or combinations thereof as a function of protein profile. This allowed us to confirm a number of known and suggest a number of novel promising drugs and targets specific to the integrated Full Proteome Subtypes identified in this project. One example of a known subtype-specific drug is cetuximab, which was more effective in FPA than in FPB or FPC. Since FPA was associated with (i.e. is similar to) the transit-amplifying (TA) subtype from the subtype model by Sadanandam *et al*

(Table S2A), our analysis confirmed previous findings on cell lines and patients where cetuximab sensitivity was recurrently observed in the TA subtype (Sadanandam *et al*, 2013). Interestingly, modelling drug sensitivity as a function of kinase protein expression also confirmed previous findings by Strimpakos *et al* (2013), as high expression of EPHA2 was associated with resistance towards cetuximab. Since the Kinobeads-based expression level of EPHA2 was significantly lower in KB/FPA than in the other subtypes, one might speculate that treatment with EPHA2-targeted drugs might re-sensitise cetuximab-resistant patients and cell lines to the antibody. Koch *et al* previously made similar observations for non-small-cell lung cancer and proposed a model in which EGFR and EPHA2 functionally interact to mediate resistance to EGFR inhibition (Koch *et al*, 2015). Target space enrichment analysis then facilitated the identification of proteins, which are recurrently targeted by drugs specific to certain FPSs. This analysis enabled the suggestion of target-based treatment options for each FPS, while the drug sensitivity predictions themselves provide more detailed information for the selection of rational treatment regimens as part of, for example, prospective clinical trials. For example, selumetinib (MEK inhibitor), lapatinib (EGFR inhibitor) and vorinostat (HDAC inhibitor) were specific to FPA, SN-38 (topoisomerase inhibitor) and sorafenib (a RAF inhibitor) were specific to FPB, while AZD7762 (checkpoint kinase inhibitor) and methotrexate (DHFR inhibitor) were specific to FPC. Together with our target space enrichment analysis, this suggested that patients of the FPA subtype might profit most from drugs targeting the classical MAPK signalling cascade or acetylation-based epigenetic modifications, patients of the FPB subtype might benefit most from drugs promoting apoptosis and patients of the FPC subtype might be best served by DHFR-targeted drugs.

The kinase-centric drug sensitivity analysis found high expression of MAP2K1 (MEK1) frequently associated with resistance towards EGFR inhibitors, and high expression of MERTK recurrently associated with resistance towards MEK1/2 inhibitors. While activating mutations of K57 in MAP2K1 are known to play a role in the development of resistance towards EGFR-targeted treatments (Bertotti *et al*, 2015), to our knowledge, MERTK was so far only shown to be up-regulated on the mRNA level in pancreatic cancer cell lines resistant towards the MEK1/2 inhibitor selumetinib (Beech & Kelly, 2014). We were able to confirm the predictive potential of MERTK protein expression for CRC cell lines *in vitro*. In addition, exploration of MERTK expression levels in tumours using immunohistochemistry on tissue microarrays (TMAs) showed that high MERTK expression is a biomarker for worse overall, disease-free and recurrence-free survival in CRC patients. Aberrant expression of MERTK was observed previously in a variety of cancers (Graham *et al*, 2014). We showed that high MERTK expression was associated with stage T4 tumours. Tavazoie *et al* (2008) suggested that MERTK is a target of miR-335-dependent post-transcriptional regulation. Since miR-335 was shown to be down-regulated in aggressive CRC and breast cancer (Tavazoie *et al*, 2008; Sun *et al*, 2014), one could speculate that aberrant expression of MERTK in CRC may be due to down-regulation of miR-335. Given that MERTK is a receptor tyrosine kinase, the protein might also be an actionable target for the treatment of advanced CRC, possibly in combination with MEK1/2 inhibitors. However, further work is needed to evaluate whether MERTK inhibition alone or combination treatments have clinical potential in CRC. In the light of the fact that, for example,

the approved drugs crizotinib and sunitinib are also potent MERTK inhibitors and that the designated compound MRX-2843 will soon enter clinical trials, such investigations are becoming feasible in future.

# Materials and Methods

### Cell lines

All cell lines in the CRC65 panel apart from SW480 (kind gift from Ulrike Stein from the MDC, Berlin) were collected by the laboratory of Prof Sir Walter Bodmer FRS at the University of Oxford and were previously HLA-typed and characterised for other genetic changes to determine whether they are derived from cancers of different donors (Emaduddin *et al*, 2008). Information on the CRC65 cell lines and the CPTAC tumour samples like MSI status, original sources (cell lines) or subtype membership is compiled in Table EV6A.

### Patient samples

We analysed the expression of MERTK in 1,074 patients from the QUASAR 2 clinical trial cohort, with approval from the West Midlands Research Ethics Committee (Edgbaston, Birmingham, UK; REC reference: 04/MRE/11/18). All participants provided written informed consent for treatment, and separate consent was obtained regarding the use of tumour tissue. QUASAR 2 is a phase III international randomised controlled trial, which collected data on toxicity, overall survival (OS), disease-free survival (DFS) and recurrence-free survival (RFS) for 1,941 stage II/III CRC patients, with the aim to determine the efficacy of adjuvant capecitabine±bevacizumab after resection of the primary tumour. A biobank comprising 1,350 FFPE blocks was established, and tissue microarrays (TMAs) were generated from 1.2-mm cores.

### Cell culture and lysis

Slightly altering the culture conditions described by Emaduddin *et al* (2008), cells were grown in high glucose Dulbecco's modified Eagle medium (DMEM, including GlutaMAX and pyruvate; PAA) containing 1% Pen-Strep (penicillin at 100 units/ml and streptomycin at 100 μg/ml final concentration, PAA) and 10% foetal bovine serum (FBS, SLI EU-000F Batch 503005) in a humidified incubator at 37°C and 10% $CO_2$ using T-175 flasks (Corning). Hereafter, this medium composition is referred to as "culture medium" and the culture conditions apart from the culture vessel are referred to as "standard culture conditions". Adherent cells were harvested at ~80–90% confluency. Suspension cells (e.g. HDC-135) growing in 250 ml of culture medium were harvested by centrifugation in 250-ml centrifuge tubes (Corning) at 300 ×*g* and 4°C for 5 min. We used RIPA100 buffer (20 mM Tris–HCl pH 7.5, 1 mM EDTA, 100 mM NaCl, 1% Triton X-100, 0.5% sodium deoxycholate and 0.1% SDS) containing protease (Complete™ mini with EDTA; Roche) and phosphatase inhibitors (Phosphatase Inhibitor cocktail 1 and 2; Sigma Aldrich) at 2× and 5× the final concentration recommended by the manufacturer, respectively, and lysed the cells for 30 min at 4°C. After clearing the total cell lysates (TCLs) at 22,000 ×*g* for 30 min at 4°C, the protein concentration was determined using a Coomassie-based

protein assay kit (Thermo Fisher) and the TCLs were stored at −80°C until further use.

## Cell viability assays

We performed *in vitro* cell viability assays in order to test our *in silico* predictions. Assays were performed as described previously (Garnett *et al*, 2012), with minor modifications. Cell viability assays were carried out in technical triplicates in order to generate 10-point dose–response curves for drugs with the resistance of which either ACVR2A or MERTK was associated. For each drug, we selected three cell lines predicted to be resistant and three cell lines predicted to be sensitive towards the respective drug and seeded them in 100 μl culture medium at their optimal seeding density on day zero (C10 = 4,000, CaCo-2 = 2,000, CC07 = 4,000, HDC-143 = 12,000, SK-CO-1 = 8,000, T84 = 11,000, RKO = 4,000, LS 180 = 2,000, HDC-111 = 4,000, HT55 = 12,000, OXCO-1 = 8,000, C125-PM = 11,000). Following an overnight incubation under standard culture conditions, 50 μl of fresh medium containing either 1% DMSO or the respective drug in a 9-point twofold dilution series in 1% DMSO was added to the corresponding wells. This resulted in a final DMSO concentration of 0.33%, while the highest final drug concentration was 10 μM for UNC569 (Merck), as well as 0.5 μM for PD-0325901, 5 μM for RDEA119, 0.5 μM for AUY922 (all from Cambridge Bioscience) and 16 μM for BAY 61-3606 (Insight Biotechnology), respectively. For drug co-treatments, the respective compounds were combined at constant ratios over the entire concentration range used for the single-agent treatments, keeping the final DMSO concentration in the assay at 0.33%. After 72 h of incubation time under standard culture conditions, the medium was either replaced with 150 μl of fresh culture medium containing 1 μM of Hoechst 33342 (Thermo Fisher #H3570) or 10 μl Alamar-Blue (Thermo Fisher #88952) added to each well (only C10$^{MERTKN^{KO}}$ and C10$^{MERTKN^{WT}}$). Cells were incubated under standard culture conditions for 1 h or 4 h, respectively, before they were either imaged using an In Cell Analyzer 6000 automated confocal microscope (GE Healthcare) with four fields of view (FOVs) per well or before AlamarBlue fluorescence was quantified using a FLUOstar Omega plate reader (BMG Labtech). The Hoechst 33342 channel of all images was subsequently analysed with Columbus v2.6.0 (PerkinElmer) using two of the built-in algorithms "B" and "C" to automatically segment nuclei. For each well, we then counted the number of nuclei satisfying standard quality control criteria in all four FOVs. Subsequently, the nuclei count or AlamarBlue fluorescence was normalised to the mean of the corresponding DMSO controls, followed by dose–response modelling and parameter extraction (Appendix Supplementary Methods).

## CRISPR/Cas9 targeting of MERTK in C10 cells

Knockout of MERTK in C10 cells was installed by CRISPR/Cas9 gene targeting as described previously (Ran *et al*, 2013). Guide RNA sequences were selected by using the CRISPR Design Tool (http://www.genome-engineering.org/crispr/?page_id=41). Three guide RNAs targeting exons 7, 8 (encoding the FNIII domain) and 14 (encoding the kinase domain) of MERTK were obtained, which were designed to induce double-strand breaks at position 982, 1,273 and 1,882 bp. The guide RNA sequences were cloned into pSpCas9

(BB)-2A-GFP (the vector was a gift from Feng Zhang; Addgene plasmid #48138) by standard Golden Gate Assembly using the BbsI site, followed by transformation of chemically competent DH5α cells. Three colonies of each sgRNA transformation were picked and the correct insertion confirmed by sequencing. C10 cells were transfected with a mixture of all three sgRNAs using Lipofectamine 3000 (Invitrogen) according to the manufacturer's instructions. Transfected cells were microscopically identified by expression of GFP and sub-cloned 3 days post-transfection using cloning rings. After 2–4 weeks, individual cell clones were tested for successful knockout of MERTK by Western blot and sequencing.

## Western blots

Total cell lysates were diluted to 1 mg/ml protein concentration and 1× final sample buffer concentration with 4× sample buffer (70 mM Tris pH 6.8, 5% v/v 2-mercaptoethanol, 40% v/v glycerol, 3% w/v SDS, 0.05% w/v bromophenol blue) and stored at −80°C until further use. Samples were separated using 4-12% NuPAGE Bis-Tris mini/midi gels (70 μg/14 μg per sample) and subsequently blotted to Hybond-P PVDF (Amersham) or nitrocellulose (iBlot Transfer Stack, Thermo Fisher) membranes according to the manufacturer's instructions. We used primary antibodies against EPHA4 (ab157588, 1:500, Abcam), ABL1 (OP20, 1:1,000, Merck), BRAF$^{V600E}$ (E19290, 1:500, Spring Bioscience) and ERK1/2 (#4695, 1:1,000, Cell Signaling Technology) for PVDF and primary antibodies against MERTK (ab52968, 1:2,000, Abcam) and ACTB (sc-47778, 1:1,000, Santa Cruz) for nitrocellulose membranes. Membranes were incubated in the dark with appropriate fluorophore-coupled secondary antibodies (IRDye 800CW goat anti-mouse IgG #926-32210, IRDye 680RD donkey anti-rabbit IgG #926-68023, IRDye 800CW goat anti-rabbit IgG #926-32211 and IRDye 680RD goat anti-mouse IgG #925-68070). Membranes were measured using an Odyssey near-infrared scanner (LI-COR). Densitometry of Western blots was carried out using ImageStudioLite v5.2.5 (LI-COR), expressing EPHA4 and ABL1 expression relative to the respective ERK1/2 signal, followed by dividing all expression values by the expression value of OXCO-1 (present on each gel) and log2-transformation.

## Immunohistochemistry

In order to evaluate MERTK expression in tissue microarrays, a staining protocol was developed using positive and negative control cell lines for MERTK expression, selected based on the expression level as measured by LC-MS/MS. For each of the four control cell lines (negative: RKO, CC07; positive: HT55, CaCo-2), we grew one T-175 flask under standard culture conditions in culture medium to ~70% confluency, washed the cells once with 10 ml ice-cold PBS and subsequently scraped them into 2 ml 4% PFA in PBS, followed by centrifugation at 400 ×*g* for 10 min and fixation overnight. Afterwards, the cell pellets were embedded in paraffin using standard procedures. Immunohistochemistry (IHC) of QUASAR 2 TMAs and FFPE control cell pellets was carried out as already described (Schlegel *et al*, 2013), with minor modifications. Stainings quantifying MERTK expression were performed using the Bond-MAX automated IHC system (Leica Biosystems) at room temperature if not specified otherwise. After deparaffinising and re-hydrating the slides, heat-induced epitope retrieval (HIER) was performed for 10 min at

100°C with epitope retrieval solution 1 (AR9961, pH 6). MERTK was detected using the Bond Polymer Refine Detection Kit (DS9800) together with a primary antibody against MERTK (ab52968, 1:1,000, Abcam) according to the manufacturer's instructions, followed by counterstaining nuclei with haematoxylin solution (< 0.1% haematoxylin). Slides were washed, dehydrated with an increasing alcohol series (30 s in 50, 70, 100 and 100% EtOH), cleared with two 30-s washes in 100% xylene and subsequently mounted using DPX mountant (Sigma Aldrich). Finally, slides were scanned at 40× magnification using a ScanScope (Aperio).

**Sample processing for mass spectrometry**

*Full proteomes*
**Acetone precipitation and re-solubilisation** TCLs were first acetone-precipitated overnight using four volumes of pre-cooled (−40°C) acetone to remove detergents, followed by two additional washing steps with 1 ml fresh, cold acetone. In between precipitation and washing steps, samples were centrifuged for 10 min at 13,000 ×g and 4°C. After the final washing step, the supernatant was taken off and the samples were left to dry in a fume hood at room temperature. Following acetone precipitation, the samples were re-suspended in urea buffer (40 mM Tris–HCl pH = 7.6, 8 M urea) containing protease (Complete mini without EDTA; Roche) and phosphatase inhibitors (Phosphatase Inhibitor cocktail 1 and 2; Sigma Aldrich) at 1× and 5× the final concentration recommended by the manufacturer, respectively, as well as 20 nM calyculin A. In order to ensure proper re-solubilisation of proteins, the precipitates were first mixed thoroughly by pipetting up and down, followed by sonication of each sample for 5 min on ice using an HF generator GM mini20, equipped with an ultrasonic converter UW mini20 and a microtip MS 2.5 sonotrode (Bandelin), which was set to 3-s pulses at 30% intensity with 3-s pause in between. After re-solubilisation, the protein concentration of the lysate was determined again using a Coomassie-based protein assay kit (Thermo Fisher).

**In-solution digestion** For in-solution digestion of proteins, 3.5 mg of TCL per sample was reduced with 10 mM DTT and subsequently alkylated using 55 mM chloroacetamide. Afterwards, samples were diluted with 40 mM Tris–HCl pH 7.6 to reduce the urea concentration to 1.5 M, 1.5 μl of $CaCl_2$ was added to each sample and proteins were digested overnight at 37°C and 700 rpm in a thermomixer using trypsin (Roche) at a protease-to-protein ratio of 1:50.

**Desalting** Desalting of peptide mixtures was carried out at room temperature using Sep-Pak cartridges (50 mg sorbent per cartridge, Waters) and a vacuum manifold according to the manufacturer's instructions. Desalted samples were stored at −80°C until further use.

**hSAX chromatography** For hydrophilic strong anion exchange (hSAX) chromatography, peptide solutions were first dried down in a Speed-Vac, re-solubilised in hSAX solvent A (5 mM Tris–HCl, pH 8.5) to a concentration of 2.73 μg/μl peptide and then centrifuged for 30 s at 5,000 ×g to spin down insoluble debris. Chromatography was carried out using a Dionex Ultimate 3000 HPLC system (Thermo Fisher), which was equipped with an IonPac AG24 guard column (Thermo Fisher), as well as an IonPac AS24 strong anion exchange column (Thermo Fisher). Chromatography was performed

at 30°C and a flow rate of 250 μl/min. Following the injection of 100 μl sample and 3 min of equilibration with 100% hSAX solvent A, peptides were eluted using a two-step linear gradient from 0 to 25% hSAX solvent B (5 mM Tris–HCl, pH 8.5, 1 M NaCl) in 24 min and from 25 to 100% solvent B in 13 min. Solvent B was kept at 100% for another 4 min to flush the column before returning to 0% in 1 min and additional 5 min of equilibration with 100% hSAX solvent A. We started collecting 48 fractions of 250 μl after 2 min of gradient time, which were subsequently combined to form 24 fractions after consulting the 214-nm chromatography trace. For that, fractions 5–7 were pooled to form fraction 5, fractions 24–25 formed fraction 22, fractions 26–28 formed fraction 23 and fractions 29–48 formed fraction 24, respectively.

**Post-hSAX desalting** All 24 fractions were desalted using StageTips as described earlier (Rappsilber *et al*, 2007), with minor modifications. Briefly, we used five C18 discs (3M Empore) of about 1.5 mm diameter in 200-μl pipette tips. After wetting (MeOH followed by 5% TFA in 80% ACN) and equilibration of stage tips (0.1% TFA), acidified samples (pH 2) were loaded twice onto the StageTips to ensure proper binding. Following two washes with 0.1% TFA, samples were eluted using 200 μl 0.1% TFA in 60% ACN. Eluates were transferred to 96-well plates, evaporated to dryness using a Speed-Vac and subsequently stored at −20°C until further use.

*Kinobeads*
Kinobeads gamma (KBγ) pulldowns (biological triplicates) were performed in 96-well filter plates (Porvair Sciences) as described elsewhere (Medard *et al*, 2015), with minor modifications. For each pulldown, 3 ml of 2.2 mg/ml TCL was cleared by ultracentrifugation at 167,000 ×g and 4°C for 20 min. Following washing (CP buffer: 50 mM Tris–HCl pH 7.5, 5% glycerol, 1.5 mM $MgCl_2$, 150 mM NaCl, 1 mM $Na_3VO_4$) and equilibration (CP buffer supplemented with 0.4% Igepal CA-630) of 35 μl settled KBγ per TCL, ~1.8 ml (equivalent of 4 mg of protein) of each TCL was transferred to its corresponding well. After 60-min incubation at 4°C on a head-over-end shaker, the beads were washed thrice with CP buffer containing 0.4% Igepal CA-630 and twice with CP buffer containing 0.2% Igepal CA-630. Subsequently, proteins were eluted by incubating the beads for 30 min at 50°C and 700 rpm in a thermomixer with 60 μl of 2× NuPAGE LDS sample buffer (Thermo Fisher) per well. After collecting eluates by centrifugation, samples were alkylated using 55 mM chloroacetamide. Finally, detergents and salts were removed from samples by running a short electrophoresis (~0.5 cm) using 4–12% NuPAGE gels (Thermo Fisher), followed by tryptic in-gel digestion according to the standard procedures.

**LC-MS/MS data acquisition**

*Full proteomes*
**Reverse-phase gradient** Full proteome fractions were measured using nanoflow LC-MS/MS by directly coupling an Eksigent nanoLC-Ultra 1D+ (Eksigent) to an Orbitrap Velos mass spectrometer (Thermo Fisher). Peptides were dissolved in 20 μl buffer A (0.1% FA), injecting 5 μl per measurement. Using a flow rate of 5 μl/min, peptides were then loaded onto a 2-cm trap column

(100 μm i.d., ReproSil-Pur 120 ODS-3 5 μm, Dr. Maisch) and washed for 10 min with 100% buffer A. Subsequently, peptides were separated on a 40-cm analytical column (75 μm i.d., ReproSil-Gold 120 C18 3 μm, Dr. Maisch) using a flow rate of 300 nl/min and a gradient from 2% buffer B (0.1% FA and 5% DMSO in ACN) to 4% in 2 min (buffer A now also contained 5% DMSO) and from 4 to 32% in 96 min. Buffer B was then ramped from 32 to 80% in 1 min and the column was flushed with 80% buffer B for 4 min, before returning to 2% buffer B in 2 min and a final equilibration step with 2% buffer B for 5 min. This resulted in a turnaround time of 120 min per full proteome fraction, totalling 130 days of measurement time for the entire CRC65 cell line panel.

**Acquisition parameters** The eluate from the analytical column was sprayed via stainless steel emitters (Thermo) at a source voltage of 2.6 kV towards the orifice of the mass spectrometer; a transfer capillary heated to 275°C. The Orbitrap Velos was set to data-dependent acquisition in positive ion mode, automatically selecting the top 10 most intense precursor ions from the preceding full MS (MS1) spectrum with an isolation width of 2.0 Th for fragmentation using higher-energy collisional dissociation (HCD) at 30% normalised collision energy and subsequent identification by MS/MS (MS2). MS1 (360–1,300 $m/z$) and MS2 (precursor-dependent $m/z$ range, starting at $m/z$ 100) spectra were acquired in the Orbitrap using a resolution of 30,000 and 7,500 at $m/z$ 400, with an automatic gain control (AGC) target value of $1 \times 10^{6}$ and $3 \times 10^{4}$ charges and a maximum injection time of 100 and 200 ms, respectively. Dynamic exclusion was set to 60 s.

*Kinobeads*

**Reverse-phase gradient** Kinobeads eluates were measured using nanoflow LC-MS/MS by directly coupling an Eksigent nanoLC-Ultra 1D+ (Eksigent) to an Orbitrap Elite mass spectrometer (Thermo Fisher). Chromatography as well as data acquisition was similar to the settings used for full proteome fractions; hence, we only describe differences between the two set-ups. We injected 10 μl per measurement instead of 5 μl and used a slightly steeper gradient from 2% buffer B to 4% in 2 min and from 4 to 32% in 88 min instead of 96 min. The rest of the gradient was kept the same, resulting in a turnaround time of 110 min per Kinobeads pulldown, totalling ~15 days of measurement time for the entire CRC65 cell line panel in biological triplicates.

**Acquisition parameters** The source voltage was at 2.2 kV instead of 2.6 kV and the Orbitrap Elite selected the top 15 most intense precursor ions for MS2 instead of the top 10 most intense ones because of its higher scanning speed. While the MS1 $m/z$ range, resolution, AGC target value and maximum injection time were the same as for the Orbitrap Velos, the Orbitrap Elite acquired MS2 spectra at a resolution of 15,000 at $m/z$ 400 with an AGC target value of $2 \times 10^{4}$ charges and a maximum injection time of 100 ms. Dynamic exclusion was set to 20 s instead of 60 s and the Orbitrap Elite also made use of a global kinase peptide inclusion list, which contained precursor ions and retention times from frequently observed kinase peptides.

## Processing of LC-MS/MS raw data

*Full proteomes and CPTAC patient data*

MaxQuant v.1.5.3.30 (Cox & Mann, 2008) was used to search our LC-MS/MS raw data, as well as the raw data from the original CPTAC publication on human colon and rectal cancer (Zhang *et al*, 2014) against the UniProtKB human reference proteome (v25.11.2015; 92,011 sequences), concatenated with a list of common contaminants supplied by MaxQuant (245 sequences) in two separate runs with identical settings. Therefore, some data used in this publication were generated by the Clinical Proteomic Tumor Analysis Consortium (NCI/NIH). We set the digestion mode to fully tryptic, allowing for cleavage before proline (Trypsin/P) and a maximum of two missed cleavages. Carbamidomethylation of cysteines was set as a fixed modification and oxidation of methionines and acetylation of protein N-termini were set as variable modifications, allowing for a maximum number of five modifications per peptide. Candidate peptides were required to have a length of at least seven amino acids, with a maximum peptide mass of 4,600 Da. The fragment ion tolerance was set to 20 ppm for FTMS (CRC65) and 0.4 Da for ITMS spectra (CPTAC), respectively. A first search with a precursor ion tolerance of 20 ppm was used to recalibrate raw data based on all peptide spectrum matches (PSMs) without filtering using hard score cut-offs. After recalibration, the data were searched with a precursor ion tolerance of 4.5 ppm, while chimeric spectra were searched a second time using MaxQuant's "Second peptides" option to identify co-fragmented peptide precursors. We used "Match between runs" with an alignment time window of 30 min and a match time window of 1.1 min to transfer identifications between raw files of the same and neighbouring fractions ($\pm$ 1 fraction). Using the classical target-decoy approach with a concatenated database of reversed peptide sequences, data were filtered using a PSM and protein false discovery rate (FDR) of 1%. Protein groups were required to have at least one unique or razor peptide, with each razor peptide being used only once during the calculation of the protein FDR. No score cut-offs were applied in addition to the target-decoy FDR.

*Kinobeads*

Raw files from triplicate Kinobeads pulldowns were processed using a pipeline similar to the one employed for the analysis of full proteomes, adapting some of the parameters described hereafter. The fragment ion tolerance was set to 120 ppm for FTMS spectra instead of 20 ppm, since we observed systematic fragment mass deviations, which were linearly dependent on the $m/z$ of the fragment ions in ppm space. Because this was likely due to problems during data acquisition, we had to compensate for it during the processing of raw data. Since Kinobeads pulldowns were not fractionated, "Match between runs" was used with the same parameters as described above to transfer identifications between all raw files.

## Quantification, statistical analysis and multi-omics data integration

Proteins detected in proteomics experiments were quantified based on MaxQuant output data, which was subsequently integrated with

transcriptomics data from various CRC samples (see Appendix Supplementary Methods for details). All statistical analyses were carried out using R v3.2.4 (R Core Team, 2016).

## Code availability

Modified MComBat and computeAUC functions (Appendix Supplementary Methods) can be downloaded from https://github.com/mfrejno/pharmacoproteomics_crc.

## Data availability

The proteomics data have been deposited to the ProteomeXchange Consortium via the PRIDE partner repository (Vizcaino *et al*, 2016) under ID codes PXD005353–PXD005355.

**Expanded View** for this article is available online.

## Acknowledgements

We thank Marcus Green and Helen Scott for help with the immunohistochemistry of TMAs, Guillaume Médard for fruitful discussions and Michaela Krötz-Fahning, Andrea Hubauer and Andreas Klaus for laboratory assistance. This work was supported in part by the Medical Research Council, the Department of Oncology of the University of Oxford and a Scatcherd European Scholarship (to M.F.).

## Author contributions

Conceptualisation: MF, SKn, SMF and BK; Methodology: MF, MW, BR, CM and SMF; Software: MF, MW and CM; Formal Analysis: MF; Investigation: MF, RZC, HK, RZ, SKl, KK, AJ, SH, MJ, JS-H and WW; Resources: EJ, ED, DK, WW, SKn, SMF and BK; Data curation: MF: Writing—original draft: MF, BK; writing—review and editing, MF, SMF and BK; Visualisation: MF; Supervision: WW, SKn, SMF and BK; Project administration: BK; Funding acquisition, MF, SKn, SMF and BK.

## Conflict of interest

The authors declare that they have no conflict of interest.

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
