## [Review Process File · Molecular Systems Biology]

Pharmacoproteomic characterisation of human colon and rectal cancer

Martin Frejno, Riccardo Zenezini Chiozzi, Mathias Wilhelm, Heiner Koch, Runsheng Zheng, Susan Klaeger, Benjamin Ruprecht, Chen Meng, Karl Kramer, Anna Jarzab, Stephanie Heinzlmeir, Elaine Johnstone, Enric Domingo, David Kerr, Moritz Jesinghaus, Julia Slotta-Huspenina, Wilko Weichert, Stefan Knapp, Stephan M Feller & Bernhard Kuster

Corresponding author: Stephan M Feller, University of Oxford & Bernhard Kuster, Technical University of Munich

Review timeline:

Submission date:	20 April 2017
Editorial Decision:	30 May 2017
Revision received:	28 August 2017
Editorial Decision:	21 September 2017
Revision received:	27 September 2017
Accepted:	28 September 2017

Editor: Maria Polychronidou

Transaction Report:

1st Editorial Decision

30 May 2017

Thank you again for submitting your work to Molecular Systems Biology. We have now heard back from two of the three referees who agreed to evaluate your study. Since their recommendations are quite similar I prefer to make a decision now rather than further delaying the process. As you will see below, the reviewers raise a number of concerns, which we would ask you to address in a revision.

The referees' recommendations are rather clear so I think that there is no need to repeat the points listed below. However, please do not hesitate to contact me if you would like to discuss any of the points in further detail.

REVIEWER REPORTS

Reviewer #1:

Frejno et al presented an integrated proteomic-transcriptomic analysis of colorectal cancer. They generated a deep proteomic dataset and the kinome of 65 colorectal cancer cell lines and integrated these data with publically available tumor data from CPTAC and multiple mRNA expression

profiles from RNA-seq and microarray experiments. Their analyses associate cell lines with specific tumor subtypes, and further predicts the sensitivity to drugs. Overall, I find the data of high quality and with potential importance to the systems- biology community. Nevertheless, I find some flaws that require revision.

1. The claim that the ratio of mRNA/protein is almost constant is not supported by the results. Figure 2A clearly shows 10-fold differences between cell lines and tissues. Presenting the data in a log₁₀ scale is simply misleading since the biologically relevant scale is often 2-5 fold, which is clearly changing according to their results. The authors also cite their own work, which suffers from the same flaw. I do not object to predicting missing values based on mRNA, but I totally disagree with the claims and presentation in figure 2A.
2. Along the same lines, it is inappropriate to refer to protein and mRNA and 'batches', use the MComBat algorithm to remove batch-effects, and then claim that protein and mRNA are the same. It is again a misleading statement, which results from very aggressive data manipulation.
3. The authors present various ways of imputations (on top of the mRNA-based imputation) in Figure EV2. First, they used wrong parameters in Perseus, which led to a bimodal distribution. These values have to be adjusted per analysis, and one cannot simply copy from a prior publication. In addition, since the authors elaborate on data imputation approaches, they should compare the different approaches, as well as no imputation at all, in terms of clustering and enrichment. I am not convinced that taking the elaborate approach actually has any benefit over the standard approaches that are commonly used (when properly done).
4. What is the benefit of the separate kinome measurement? Given the great depth of their proteome measurements, I can speculate that almost all kinases identified in the kinome are also seen in the proteome. They should present clearly the benefit of these measurements and compare the quantitative differences seen between these experimental approaches.
5. The authors use multiple levels of data normalization/manipulation and eventually it is confusing which result originates from which type of normalization. The authors should clearly state, for each analysis/figure which type of data was used (gIbaq, imputation type, LFQ etc).
6. The background of the enrichment analyses is not clearly stated. They should use the identified proteome as background and not the entire genome/proteome, to eliminate data acquisition biases.
7. Figure 1B should be better explained. The authors claim high degree of agreement between the known subtypes and the prediction. However it seems like only CMS2 is well predicted, but the other three look quite mixed. They should provide statistical measure of their claim of good prediction, and separate between subtypes in their analyses.
8. Figure 3 shows total mismatch between the FP subtypes and CMS, which does not agree with their previous claims of high concordance between the data layers and high prediction ability. This result is actually more convincing than the others, and should be elaborated.

Minor comments:

1. Western blots in figure EV4 have to be improved (especially for Erk).
2. Method section indicates that all MSMS were done in the Orbitrap, but they set 0.4Da mass tolerance for ITMS. Should be corrected.
3. Figure EV1- Y axis range should fit the measured data.
4. Tables of identified proteins should include quality measures, such as q-value/PEP, # peptides, etc.
5. Figure 5C bottom graphs seem to contradict their claim. Overall, they should add statistical evaluation of the difference between resistant and sensitive cells.

Reviewer #3:

In this study, the authors first performed proteomic profiling of 65 CRC cell lines and quantified expressed kinomes. By combining mRNA data sets and proteomic data sets, the authors classified CRC cell lines and patient derived data into CMSs and demonstrated good concordance between mRNA and protein expression, and merged cell line and patient datasets into one protein expression matrix. They identified three subgroups with distinct characteristics (FPA, FPB and FPC) and resulting in protein signatures that predict drug sensitivity. The authors used consensus clustering to identify three Kinobeads subtypes (KA, KB and KC), and proposed that MERTK expression can be used to predict resistance to MEK1/2 inhibitors. They also quantified MERTK abundance by IHC in

tumor samples from the QUASAR 2 trial and proposed that MERTK is a prognostic survival marker in CRC patients.

Comments:

This manuscript is overall well written and convincing. The integration of CRC cell line and patient proteomic data with mRNA data resulted in a new classification of CRC subtypes with different drug sensitivities. This approach could potentially be applied to predict drug sensitivity for other cancer types. The lists of proteome and kinome profiles as well as predicted drug sensitivities are useful resources for other researchers.

QUASAR 2 is a study comparing the effect of treatment using capecitabine to capecitabine with bevacizumab on survival. The authors should perform survival analysis based on groups treated with or without bevacizumab.

The authors claim that "MERTK expression was found primarily at the membrane, with some expression observed in the cytoplasm" with typical surface staining shown for Caco2 cells by IHC. However, in figure 6C, there appears to be mainly cytosolic staining rather than membrane staining. It may be useful to quantify the intensity of MERTK in the surface and cytosol compartments separately, and analyze both with survival data. Also, in proposing that MERTK is a therapeutic target, in addition to testing drug sensitivity to MEK1/2 inhibitors in MERTK high and low cells, the authors should also investigate knockdown or overexpression, followed by drug treatment to avoid the complication of different genetic backgrounds and directly associate MERTK to MEK1/2 inhibitor drug sensitivity.

Minor points

1. In the abstract, the authors should indicate "1,074 CRC tumors" instead of "1,000"
2. In row 112, please clarify where the 81 patients are from, as there are 89 tumor samples in table EV1D.
3. The paper needs to be corrected for typos

1st Revision - authors' response

28 August 2017

Pharmacoproteomic characterisation of human colon and rectal cancer

Detailed response to reviewer comments:

The authors are grateful to the comments made by the reviewers. The new data, data analysis, figures and text have made the manuscript much stronger and the authors hope that all concerns have been adequately addressed. Specific details are provided below but the most significant changes to the manuscript are:

- *We moved main Figure 2 to the supplement and exchanged it with the old Figure EV1.*
- *We repeated the western blots for ABL, EPHA4, ERK1/2 and BRAF^{V600E} for all 65 cell lines.*
- *We created a CRISPR/CAS9 knockout cell line for MERTK to validate observed drug sensitivity effects.*

Reviewer #1

Frejno et al presented an integrated proteomic-transcriptomic analysis of colorectal cancer. They generated a deep proteomic dataset and the kinome of 65 colorectal cancer cell lines and integrated these data with publically available tumor data from CPTAC and multiple mRNA expression profiles from RNA-seq and microarray experiments. Their analyses associate cell lines with specific tumor subtypes, and further predicts the sensitivity to drugs. Overall, I find the data of high quality and with potential importance to the systems- biology community. Nevertheless, I find some flaws that require revision.

1. The claim that the ratio of mRNA/protein is almost constant is not supported by the results. Figure 2A clearly shows 10-fold differences between cell lines and tissues. Presenting the data in a log10 scale is simply misleading since the biologically relevant scale is often 2-5 fold, which is clearly changing according to their results. The authors also cite their own work, which suffers from

the same flaw. I do not object to predicting missing values based on mRNA, but I totally disagree with the claims and presentation in figure 2A.

Our original text may have given the wrong impression of what we were trying to accomplish. We have clarified in the revised text that the protein/mRNA ratios are only used for the purpose of dealing with missing values in the proteomics data. The reason we showed the four examples on a log10 scale was merely for visualisation because they illustrate examples of very different protein/mRNA ratios, which is impossible to show on an absolute scale. We have also moved Figure 2 into the supplement as it may distract the reader from the main story line. To further clarify, we do not claim that protein and mRNA measurements are the same, as we are well aware of the fact that we removed systematic differences between these measurements using MComBat. We are also not making any point about what the observed correlations of protein/mRNA levels may mean (i.e. with reference to the reviewer's point about our cited work). In the following, we provide more detail on the reviewers concern.

First, while the protein/mRNA ratios are indeed reasonably stable across samples within the CRC65 and CPTAC datasets (median absolute fold-change below 1.8 for both datasets; see Figure 2B), they are not stable across the two datasets. In fact, they are systematically higher in the CRC65 dataset compared to the CPTAC dataset, which is likely due to a combination of factors. Two important ones are the difference in the amount of digest used for the analysis and the depth of peptide fractionation. The CRC65 analysis consumed about 68 µg of digest per cell line (8 µg for CPTAC) and separated the peptides into 24 fractions (15 for CPTAC), resulting in substantially more peptides per protein in the CRC65 compared to the CPTAC dataset (Figure 2 of the revised manuscript and new panel in Figure EV2). As a result, iBAQ protein intensities (and also giBAQ intensities) are not well comparable between the datasets because iBAQ values are calculated by summing up individual peptide intensities and subsequently dividing them by the number of theoretically observable peptides per protein/gene. Therefore, protein/mRNA ratios calculated on two datasets with incomparable proteomic "depth" are bound to differ for the majority of all proteins, with the expectation to observe smaller ratios in the "shallower" dataset. This is exactly what we observe in Figure 2A if we compare for example the median protein/mRNA ratio of GAPDH in the two datasets. In part, this could have been addressed by total-sum-normalisation of each dataset (see original Figures EV2C and D). However, we refrained from doing so, since total sum normalisation only accounts for global differences between the datasets and implicitly assumes that the datasets show similar peptide coverage for all proteins. Because this is not the case, we rather decided to correct for these differences in a protein-by-protein fashion using ComBat as described in the main manuscript. To verify this approach, we compared total sum normalisation with ComBat and found that total sum normalisation clustered all cell lines (or tumours) while ComBat processed data was able to generate clusters that contain both cell lines and tumours. Second, regarding the variation of protein/mRNA ratios we note (and depicted in original Figure 2B) that the variability of protein/mRNA ratios or fold-changes for the majority of all proteins is smaller than the aforementioned 2-5 fold. The median absolute fold-change is below 1.8 for each dataset, which is within the expected measurement (in)accuracy of label-free proteomics experiments. Furthermore, 90% of all protein/mRNA fold-changes deviate from their corresponding median fold-change by less than 5-fold and 62% deviate by less than 2 fold. We acknowledge the fact that (original) Figure 2A shows much larger protein/mRNA fold-changes in some samples, (as much as 7.7-fold for PPIA in sample 8H). This is particularly the case for the CPTAC dataset. However, this is only observed for a few tumour samples and might be explained by inconsistencies during sample preparation or differences in tumour cellularity. In fact, sample 3A visually shows much higher deviation of its protein/mRNA fold-changes relative to the corresponding median compared to all other tumour samples, suggesting that 3A is an outlier sample.

2. Along the same lines, it is inappropriate to refer to protein and mRNA and 'batches', use the MComBat algorithm to remove batch effects, and then claim that protein and mRNA are the same. It is again a misleading statement, which results from very aggressive data manipulation.

Please also see our comments above. We agree that the term 'batches' might not be appropriate in this context and we thus replaced it by "systematic differences". Our analysis clearly showed systematic differences between mRNA and protein measurements, and the removal of these systematic differences in a gene-by-gene fashion was required to enable mRNA-guided missing value imputation (see Expanded View Appendix). For this purpose, we used MComBat, which is designed to accomplish exactly that. Given that the reviewer did not object to impute missing values

in proteomics data based on mRNA measurements, we did not modify the method itself but added some text to the main manuscript and Expanded View Appendix to make this point clearer. We would like to emphasise again that we never intended to imply that protein and mRNA measurements are the same. Throughout the manuscript, we clearly stated that a high correlation of protein and mRNA measurements was only observed after removal of systematic differences between them.

3. The authors present various ways of imputations (on top of the mRNA-based imputation) in Figure EV2. First, they used wrong parameters in Perseus, which led to a bimodal distribution. These values have to be adjusted per analysis, and one cannot simply copy from a prior publication. In addition, since the authors elaborate on data imputation approaches, they should compare the different approaches, as well as no imputation at all, in terms of clustering and enrichment. I am not convinced that taking the elaborate approach actually has any benefit over the standard approaches that are commonly used (when properly done).

We had specifically looked at this point but did not include a discussion in the manuscript because the results were rather clear. Our main concern was neither the clustering, nor the enrichment analysis, but the elastic net regression which requires a complete data matrix and for which the intensities of individual proteins (rather than the overall distribution of proteins) is of great importance. If no imputation was performed, the clustering did not work properly (i.e. merely clustering the samples according to the two groups of cell lines and tumours). When adjusting the parameters of the perseus-type imputation such that the imputed values became more and more (eventually fully) part of the overall distribution, we found an ever increasing and eventually large number of cases in which the imputed values for specific proteins had much higher intensities than the same but experimentally robustly measured protein in a different sample. This clearly introduces substantial error, which we were able to avoid when using our approach of minimum-guided missing value imputation. While using default perseus-type settings and our method resulted in essentially the same outcome at the level of clustering, we chose 'our' imputation method because it is more suitable for the subsequent elastic net analysis, since it ensures that the imputed abundance of a given protein is never higher than the minimal measured abundance of said protein across all samples in the respective dataset. Furthermore, the clustering ($p < 0.0005$, two-sided Fisher's Exact Test) and the fold-changes of significantly differentially expressed proteins between the detected subtypes were nearly identical when using either algorithm (Pearson's $R = 0.997$).

4. What is the benefit of the separate kinome measurement? Given the great depth of their proteome measurements, I can speculate that almost all kinases identified in the kinome are also seen in the proteome. They should present clearly the benefit of these measurements and compare the quantitative differences seen between these experimental approaches.

The benefit of the separate kinome measurement is that it enabled us to cover 55 kinases that were not present in the full proteome data (new panel in original Figure EV4) including MERTK, which we chose for further work. In addition, most kinases were also much more robustly measured using Kinobeads (because a lot more protein was used for these affinity purifications), thus enhancing the data quality for this group of proteins, which are frequent targets in oncology.

5. The authors use multiple levels of data normalization/manipulation and eventually it is confusing which result originates from which type of normalization. The authors should clearly state, for each analysis/figure which type of data was used (gIbaq, imputation type, LFQ etc).

We apologize for the lack of clarity. We have added a further Expanded View Figure (now Figure EV1) that shows a flow chart of data processing and points out which result was obtained from which type of data.

6. The background of the enrichment analyses is not clearly stated. They should use the identified proteome as background and not the entire genome/proteome, to eliminate data acquisition biases.

The authors apologize for the lack of clarity. We did exactly what the reviewer proposed. This has now been made clearer in the revised manuscript.

7. Figure 1B should be better explained. The authors claim high degree of agreement between the known subtypes and the prediction. However it seems like only CMS2 is well predicted, but the other three look quite mixed. They should provide statistical measure of their claim of good prediction, and separate between subtypes in their analyses.

Another point which we are now making clearer in the revised manuscript. First, the prediction accuracy is actually not depicted in Figure 1B. The dendrogram in the centre of the circos plot merely shows the clustering of samples from both the CRC65 and CPTAC datasets according to their mRNA expression profiles (see also Expanded View Appendix). The four clusters were originally coloured according to the CMS subtype, with which most (but not all) of the samples in a given cluster were annotated. This annotation, however, was not a result of the clustering itself, but is rather based on the single-sample predictor published by Guinney et al., which we modified so that it accepts gene symbols as identifiers rather than Entrez IDs. This is an algorithm, which takes the gene expression profile of e.g. a cell line as an input and generates the most likely CMS of said cell line as an output. We understand that this may have led to confusion, which is why we removed the colouring of the dendrogram in Figure 1B. Second, we mention a good overall prediction accuracy of 80% in the main manuscript. To substantiate/document this statement, we added Table EV2E, which contains a confusion matrix, as well as a table quantifying the prediction accuracy and precision of our modified single-sample predictor compared to the original annotation from the study by Guinney et al. for each subtype separately using a variety of commonly used metrics.

8. Figure 3 shows total mismatch between the FP subtypes and CMS, which does not agree with their previous claims of high concordance between the data layers and high prediction ability. This result is actually more convincing than the others, and should be elaborated.

Since plots like the annotation shown on top of the heat maps in Figure 3 are visually difficult to interpret, we already included statistical measures quantifying “match” and “mismatch” between the different sample annotations in the original manuscript (Table EV2A for our FP subtypes and in Table EV2C for our KS subtypes). While we do not find perfect concordance, the data does not show total mismatch between FPSs and CMSs, but rather reveals that FPA is significantly associated (concordant) with CMS2, FPB is significantly associated with CMS1/CMS4 and FBC is significantly associated with CMS3. This point may have been overlooked but we already elaborated on this in the original manuscript in the result section on ‘Integrated proteomic subtypes of CRC cell lines and tumours’, as well as in the Discussion.

Minor comments:

1. Western blots in figure EV4 have to be improved (especially for Erk).

We repeated the western blots for all four proteins and in all 65 cell lines (original Figure EV4C-D). We also switched to commercial gels and antibodies carrying chromophores for near-infrared detection in order to improve the quality of the blots.

2. Method section indicates that all MSMS were done in the Orbitrap, but they set 0.4Da mass tolerance for ITMS. Should be corrected.

We clarified this in the revised manuscript. The CPTAC data (ion trap for MS2 detection) used a 0.4 Da tolerance and the CRC65 data (Orbitrap for MS2 detection) used a 20 ppm tolerance for database searching.

3. Figure EV1- Y axis range should fit the measured data.

This was changed as suggested.

4. Tables of identified proteins should include quality measures, such as q-value/PEP, # peptides, etc.

All MaxQuant output files including the aforementioned quality metrics were uploaded to PRIDE for the original submission. We now also added all QC metrics provided by MaxQuant to table EV1A-C. We note that because this table lists gene names not protein groups, there can be more than one set of QC metrics per gene.

5. Figure 5C bottom graphs seem to contradict their claim. Overall, they should add statistical evaluation of the difference between resistant and sensitive cells.

We have removed the graphs for SN-38 and OSU-03012 from the figure (and text) in order to make room for additional data produced during the revision process upon request by both reviewers. In addition, we included the results of one-sided Mann-Whitney tests evaluating the significance of the difference between resistant and sensitive cell lines in Figure 5D. The same statistical evaluation was also included with (original) Figure EV5B and EV5F.

Reviewer #3

In this study, the authors first performed proteomic profiling of 65 CRC cell lines and quantified expressed kinomes. By combining mRNA data sets and proteomic data sets, the authors classified CRC cell lines and patient derived data into CMSs and demonstrated good concordance between mRNA and protein expression, and merged cell line and patient datasets into one protein expression matrix. They identified three subgroups with distinct characteristics (FPA, FPB and FPC) and resulting in protein signatures that predict drug sensitivity. The authors used consensus clustering to identify three Kinobeats subtypes (KA, KB and KC), and proposed that MERTK expression can be used to predict resistance to MEK1/2 inhibitors. They also quantified MERTK abundance by IHC in tumor samples from the QUASAR 2 trial and proposed that MERTK is a prognostic survival marker in CRC patients.

Comments:

This manuscript is overall well written and convincing. The integration of CRC cell line and patient proteomic data with mRNA data resulted in a new classification of CRC subtypes with different drug sensitivities. This approach could potentially be applied to predict drug sensitivity for other cancer types. The lists of proteome and kinome profiles as well as predicted drug sensitivities are useful resources for other researchers.

QUASAR 2 is a study comparing the effect of treatment using capecitabine to capecitabine with bevacizumab on survival. The authors should perform survival analysis based on groups treated with or without bevacizumab.

The authors are happy to learn that the reviewer thinks the manuscript is overall convincing and useful. Indeed, we performed univariate and multivariate survival analysis on a combination of clinical parameters (see Table EV5), which also included the treatment status of the patients (capecitabine or CAP versus capecitabine±bevacizumab or CAPBEV). For our multivariate analysis, we selected only significant variables from our univariate Cox proportional hazards regression, including the treatment status. We performed 'stepwise backward model selection starting with the full model and removing explanatory variables in order to minimise Akaike's Information Criterion (AIC)' (see Expanded View Appendix). This analysis removes insignificant predictors from the full model in order to arrive at a minimal model sufficient to explain the observed survival data. The full model included the treatment status of the patients, however the corresponding coefficient was not significant while analysing OS ($p>0.33$), DFS ($p>0.11$) and RFS ($p>0.13$). Hence, the treatment status of the patients was not included in our final multivariate Cox proportional hazards model, as it was less informative than other variables such as the tumour stage or MERTK expression. We added a sentence to the paragraph entitled 'MERTK is a prognostic survival marker in CRC patients' of the Results section mentioning this observation.

The authors claim that "MERTK expression was found primarily at the membrane, with some expression observed in the cytoplasm" with typical surface staining shown for Caco2 cells by IHC. However, in figure 6C, there appears to be mainly cytosolic staining rather than membrane staining. It may be useful to quantify the intensity of MERTK in the surface and cytosol compartments separately, and analyze both with survival data.

To clarify, our remark that that MERTK expression was found primarily at the membrane only applies to CaCo-2 cells. For the TMA stains of the patient tumours it was difficult to tell whether tumour cells with cytoplasmic staining were also positive at the membrane. Therefore, we decided to quantify MERTK expression as 'cytoplasmic/membranous'. This means that we cannot analyse

MERTK expression in these two compartments separately from one another. We amended the manuscript to clarify this point.

Also, in proposing that MERTK is a therapeutic target, in addition to testing drug sensitivity to MEK1/2 inhibitors in MERTK high and low cells, the authors should also investigate knockdown or overexpression, followed by drug treatment to avoid the complication of different genetic backgrounds and directly associate MERTK to MEK1/2 inhibitor drug sensitivity.

In order to exclude an effect of different genetic backgrounds on MEK1/2 inhibitor sensitivity, we used the CRISPR/Cas9 system to knockout MERTK in C10 cells, which usually express the protein at high levels (see Figure EV5). The western blot confirming the successful knockout of MERTK is now shown in Figure 5E. We then treated the parental cell line and the MERTK knockout clone with the same 9 concentrations of RDEA119 as done for (original) Figure 5C (upper panel). The result of this experiment is now depicted in Figure 5F and shows a moderate increase in sensitivity to RDEA119 of the knockout relative to the parental cell line. This suggests that high MERTK expression at least in part explains the resistance of C10 cells to inhibition of MEK1/2.

Minor points

1. In the abstract, the authors should indicate "1,074 CRC tumors" instead of "1,000"

This was changed according to the reviewer's suggestion.

2. In row 112, please clarify where the 81 patients are from, as there are 89 tumor samples in table EV1D.

We clarified this in the main manuscript. There are indeed 89 tumor samples in table EV1D but CMS annotations were only available for 81 samples (provided by Guinney et al.). Since reviewer #1 also suggested performing this analysis separately for each subtype, we now amended this part of the main manuscript and added Table EV2E (see also, response to reviewer #1 above), in order to provide an overview of our predictions in relation to the CMS labels published by Guinney et al.

3. The paper needs to be corrected for typos

The authors corrected all typos to the best of their knowledge as suggested by the reviewer.

2nd Editorial Decision

21 September 2017

Thank you again for sending us your revised manuscript. We have now heard back from reviewer #1 who was asked to evaluate your study. S/he is satisfied with the modifications made and thinks that the study is now suitable for publication.

Before we formally accept the study for publication, we would ask you to address a few pending minor editorial issues listed below.

REVIEWER REPORT

Reviewer #1:

The authors and addressed all my previous concerns and modified the paper accordingly. I think with these changes, they de-emphasize several problematic statements that were misleading, and their reply explain some of the analytical choices.

Based on these changes I believe the paper is much improved and I find it suitable for publication.

YOU MUST COMPLETE ALL CELLS WITH A PINK BACKGROUND ↓
PLEASE NOTE THAT THIS CHECKLIST WILL BE PUBLISHED ALONGSIDE YOUR PAPER

Corresponding Author Name: Bernhard Kuster, Stephan M Feller
Journal Submitted to: Molecular Systems Biology
Manuscript Number: MSB-17-7701